EMBO
Molecular Medicine

# ctDNA monitoring using tumor-informed copy number analysis

Ze Zhou [1,2,3], Ros Cutts[4], Sarah Hrebien[4], Christina X Zhang[2,3], Isaac Garcia-Murillas[4], Woody Z Zhang[5], Alexander M Frankell[5], Wendy N Cooper [1,2,3], Amit Roshan [1,2,3,6], Nicholas C Turner[4,7], Tommy Kaplan [1,8,9], Nitzan Rosenfeld [1,2,3✉] & Hui Zhao [1,2,3✉]

## Abstract

Methods to detect circulating tumor DNA (ctDNA) enable minimally invasive responsive monitoring of cancer dynamics. However, sensitive and cost-effective methods are still lacking. Current methods for detecting cancer signals in shallow whole-genome sequencing (sWGS) data from cell-free DNA (cfDNA) via copy number aberration (CNA) analysis typically have a limit of detection of approximately 3% tumor fraction (TF). We developed informCNA, a bioinformatics method that leverages CNA information from sWGS of tumor or pre-treatment plasma samples with high TF as references, enabling ctDNA detection down to 0.2% TF across multiple cancer types. In 177 serial plasma samples from 18 patients with ovarian cancer, informCNA showed high concordance with the standard serum protein marker CA-125 and identified recurrence a median of 3.7 months earlier than CA-125 test. These results demonstrate the potential of personalized CNA analysis through sWGS for estimating ctDNA burden, enabling precise and cost-effective disease monitoring and early detection of relapse.

**Keywords** cfDNA; ctDNA; Liquid Biopsy; Tumor-informed; Copy Number Aberration (CNA)
**Subject Categories** Biomarkers; Cancer; Methods & Resources

## Introduction

The analysis of circulating tumor-derived DNA (ctDNA) in liquid biopsies offers a minimally invasive approach to detect and monitor disease progression in cancer patients (Pantel and Alix-Panabières, 2025). The concentration of ctDNA is associated with tumor burden (Kirchweger et al, 2022), response to therapy (Tie et al, 2015) and prognosis (Gale et al, 2022). ctDNA levels are often high in preoperative or pre-treatment plasma samples (also referred to as baseline), decrease in response to effective treatment, and increase with tumor recurrence or the development of therapy resistance (Fig. 1A) (Moding et al, 2021). Monitoring ctDNA dynamics during treatment enables early intervention or therapy switching in response to molecular relapse (Bartolomucci et al, 2025).

Multiple methods, including both tumor-naive and tumor-informed assays (Santonja et al, 2023), have been developed to detect ctDNA by analyzing single-nucleotide variants (SNVs) (Wan et al, 2020; Newman et al, 2014; Zviran et al, 2020; Cohen et al, 2021), somatic copy number aberrations (CNAs) (Mouliere et al, 2018; Adalsteinsson et al, 2017) and structural variants (SVs) (Elliott et al, 2025; Santonja et al, 2023). The high prevalence of CNA in cancer (Beroukhim et al, 2010; Steele et al, 2022; Taylor et al, 2018; Shendure and Akey, 2015), makes it a valuable biomarker for cancer diagnosis and prognosis. Analysis of CNA from shallow whole-genome sequencing (sWGS) of cell-free DNA (cfDNA) offers a cost-effective option for frequent monitoring of post-treatment ctDNA dynamics compared with methods requiring deep WGS or panel-based sequencing.

Previous methods for CNA analysis in cfDNA were developed in a tumor-naive setting. Without prior information from sequencing patient-matched tumor or other samples with high tumor fraction (TF), such as pre-treatment plasma samples, these approaches have limited sensitivity for ctDNA detection (Fig. 1A). For instance, ichorCNA, one of the most widely used tumor-naive tools, employs a hidden Markov model to predict CNA segments and a Bayesian mixture model to estimate TF. It reports a lower limit of ctDNA detection (LLOD) of 3% TF with 95% sensitivity at 91% specificity, or 10% TF with 91% sensitivity at 100% specificity (Adalsteinsson et al, 2017). t-MAD, a tool previously developed by our team, detects tumor DNA by quantifying the denoised median absolute deviation (MAD) of genome-wide copy number distribution relative to the copy-neutral state. It has a proposed LLOD of 1.5% TF, based on the maximal value for samples of healthy individuals (Mouliere et al, 2018). In samples with low TF, tumor-naive approaches may also struggle to accurately estimate ploidy, resulting in biased TF estimates. Focal CNAs with high-level copy

[1]Centre for Cancer Cell and Molecular Biology, Barts Cancer Institute, Queen Mary University of London, London, UK. [2]Cancer Research UK Cambridge Institute, University of Cambridge, Li Ka Shing Centre, Cambridge, UK. [3]Cancer Research UK Cambridge Centre, Cancer Research UK Cambridge Institute, Li Ka Shing Centre, Cambridge, UK. [4]Breast Cancer Now Toby Robins Research Centre, The Institute of Cancer Research, London, UK. [5]Early Cancer Institute, University of Cambridge, Cambridge, UK. [6]Department of Plastic & Reconstructive Surgery, Cambridge University Hospitals NHS Foundation Trust, Cambridge, UK. [7]Breast Unit, The Royal Marsden Hospital, London, UK. [8]School of Computer Science and Engineering, The Hebrew University of Jerusalem, Jerusalem, Israel. [9]Department of Developmental Biology and Cancer Research, Faculty of Medicine, The Hebrew University of Jerusalem, Jerusalem, Israel. ✉E-mail: n.rosenfeld@qmul.ac.uk; huizhao@qmul.ac.uk

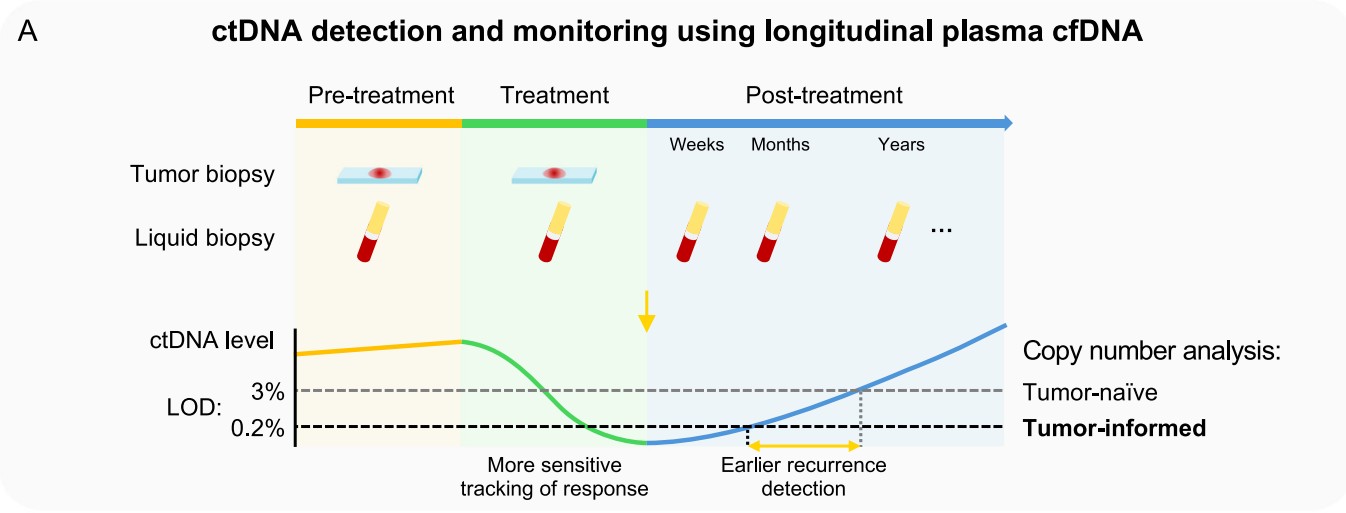

◀

**Figure 1. Schematic illustration of tumor-informed cfDNA copy number aberration (CNA) analysis pipeline, informCNA.**

(A) Tumor biopsy or liquid biopsy samples can be collected at pre-treatment (baseline), during treatment, and post-treatment time points. In the tumor-informed setting, biopsies with high tumor fraction (TF), including tumor tissue and plasma samples at baseline, can serve as reference samples to enhance ctDNA detection in follow-up query samples with low TF. (B) informCNA leverages patient-specific copy number information from the high-TF reference sample by projecting segments onto the corresponding regions of the matched query sample with low TF. It performs Copy Number Similarity Analysis to estimate TF in the query liquid biopsy samples, and conducts a complementary Copy Number Significance Analysis for ctDNA detection. informCNA achieves a lower limit of detection (LLOD) as low as 0.2% ctDNA fraction (TF). This tumor-informed approach enables earlier recurrence detection compared to tumor-naive assays. (C) informCNA was developed and validated for monitoring ctDNA dynamics in shallow whole-genome sequencing (sWGS) of plasma samples across multiple cancer types. The datasets include 30 plasma samples from three stage IV breast cancer patients (Santonja et al, 2023), 12 in vitro diluted plasma samples from two breast cancer patients, 25 plasma samples from five stage IV melanoma patients (Wan et al, 2020), and 177 plasma samples from 18 stage III-IV ovarian cancer patients (Paracchini et al, 2021).

number changes, which are critical in tumor initiation and evolution, are often missed and therefore require additional analysis using other tools (Mermel et al, 2011). In addition, the presence of aneuploid cells in normal tissue may confound these methods (Lin et al, 2024).

Tumor-informed assays based on tumor-derived SNVs have become increasingly powerful, with some demonstrating LLODs down to a few parts per million (ppm) (Santonja et al, 2023; Bae et al, 2023; Newman et al, 2016; Black et al, 2025a). However, these approaches require high-depth sequencing of tumor tissue or biopsy samples to identify patient-specific mutation lists, and most current assays require customized panel design to target these mutations (Wan et al, 2020). Tumor-informed CNA-based methods, such as MRDetect-CNA and MRD-EDGE$^{CNV}$ (Zviran et al, 2020; Widman et al, 2024), rely on deeply sequenced tumor tissue and buffy coat samples and have demonstrated the ability to detect ctDNA levels as low as 50 ppm with ~75% sensitivity at ~75% specificity (Zviran et al, 2020; Widman et al, 2024). However, these previously described tumor-informed methods (both SNV- and CNA-based) require at least 30x WGS of tumor tissue, buffy coat and plasma cfDNA, which greatly increases costs and analysis complexity. Although these costly assays, when used as single postoperative time points, can precisely identify minimal residual disease (MRD) and enable risk stratification (Zviran et al, 2020; Widman et al, 2024), thereby identifying ctDNA-negative patients who may not need additional adjuvant treatment (Tie et al, 2022), there remains a lack of accurate and cost-efficient assays for ctDNA-positive patients that enable frequent monitoring of treatment response, early detection of recurrence, and timely therapeutic decision-making.

Here, we propose a strategy that integrates the strengths of the aforementioned assays: the high sensitivity and specificity of tumor-informed assays, the broad applicability of WGS without customized panel design, and the cost-efficiency of sWGS. We hypothesized that prior knowledge of patient-specific CNAs, which can be efficiently identified at low cost from sWGS data of high-TF samples such as tumor biopsies or baseline plasma samples from cancer patients (Fig. 1B) (Adalsteinsson et al, 2017; Poell et al, 2019; Sauer et al, 2021), could be leveraged to create a cost-effective and broadly applicable approach for guiding and improving the detection and quantification of CNA-derived signals in sWGS of follow-up plasma samples, which typically have lower TFs. To this end, we developed informCNA, a tumor-informed CNA analysis method that detects ctDNA and quantifies TF using sWGS of cfDNA, without requiring a matched buffy coat sample. The method leverages patient-specific CNAs identified from high-TF reference samples (tumor biopsies or baseline liquid biopsies), then

projects these prioritized CNA segments onto matched query cfDNA samples, i.e., follow-up liquid biopsies with low TF (Fig. 1B). It estimates the ctDNA fraction in each low-TF query sample by comparing its copy number profile with that of the high-TF reference sample.

We assessed the performance of informCNA using in silico dilutions of cancer cell line DNA data and in vitro dilutions of patient ctDNA molecules into cfDNA from healthy donors. We benchmarked informCNA against three well-established copy-number–based approaches for ctDNA monitoring in patients with breast cancer. We further evaluated its clinical utility, guided by sequencing data from pre-treatment tumor tissue, for tracking disease progression and enabling earlier detection of recurrence in patients with ovarian cancer. Finally, we demonstrated the feasibility of using high-TF baseline plasma samples to guide ctDNA monitoring, highlighting the potential of non-invasive, plasma-guided informCNA for clinical context in which tumor tissue biopsy is unavailable (Fig. 1C).

## Results

### Genome-wide Copy Number Similarity Analysis enables ctDNA fraction estimation

To address the challenge of estimating the TF in query samples which may have low ctDNA levels (e.g., TF < 1%), we leveraged prior knowledge of patient-specific CNAs and TF from a high-TF (reference) sample. We quantified the extent of tumor DNA shared between reference and query samples using "Copy Number Similarity Analysis" (Fig. 2A, Methods). The resulting copy number "Similarity Score" measures how closely the query sample's copy number pattern aligns with that of the reference sample across bin-level sequencing depths spanning the genome. In brief, Copy Number Similarity Analysis employs a weighted least squares algorithm to quantify the degree of similarity between the genome-wide copy number profiles of reference and query samples, assigning additional weights to focal CNAs (Methods), which would otherwise be underrepresented due to their limited size (Beroukhim et al, 2010). The Similarity Score reflects the proportion of tumor-derived DNA shared between the reference and query samples. The TF in the query sample is then inferred as the product of the Similarity Score and the TF of the reference sample.

We first assessed the robustness of the Similarity Score using samples expected to have 100% similarity, i.e., DNA derived from the same tumor sample but sequenced at varying depths. We

**A**

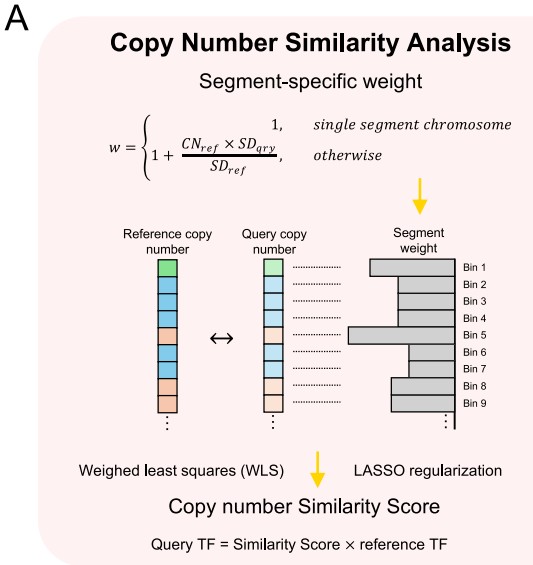

**Copy Number Similarity Analysis**

Segment-specific weight

$$w = \begin{cases} 1, & \text{single segment chromosome} \\ 1 + \dfrac{CN_{ref} \times SD_{qry}}{SD_{ref}}, & \text{otherwise} \end{cases}$$

Reference copy number ↔ Query copy number — Segment weight

Bin 1
Bin 2
Bin 3
Bin 4
Bin 5
Bin 6
Bin 7
Bin 8
Bin 9

Weighed least squares (WLS) → LASSO regularization

Copy number Similarity Score

Query TF = Similarity Score × reference TF

**B**

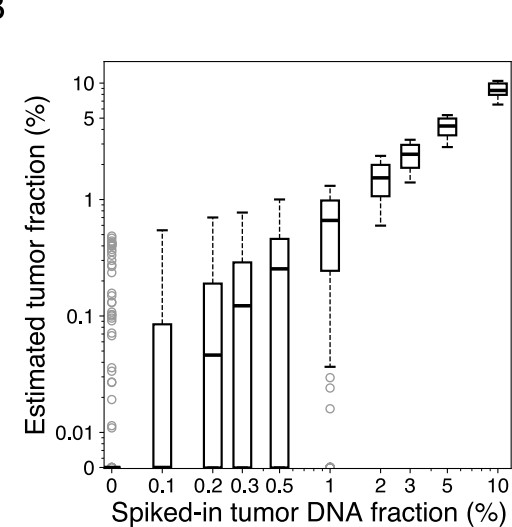

**C**

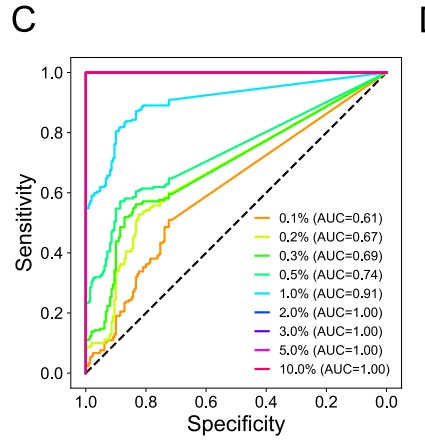

0.1% (AUC=0.61)
0.2% (AUC=0.67)
0.3% (AUC=0.69)
0.5% (AUC=0.74)
1.0% (AUC=0.91)
2.0% (AUC=1.00)
3.0% (AUC=1.00)
5.0% (AUC=1.00)
10.0% (AUC=1.00)

**D**

| TF (%): | 0.1 | 0.2 | 0.3 | 0.5 | 1 | 2 | 3 | 5 | 10 |
|---|---|---|---|---|---|---|---|---|---|
| Sensitivity at 95% specificity | 7% | 10% | 14% | 32% | 60% | 100% | 100% | 100% | 100% |
| Sensitivity at 90% specificity | 13% | 24% | 34% | 50% | 76% | 100% | 100% | 100% | 100% |
| Sensitivity at 85% specificity | 24% | 40% | 52% | 58% | 84% | 100% | 100% | 100% | 100% |
| Sensitivity at 80% specificity | 35% | 54% | 57% | 61% | 89% | 100% | 100% | 100% | 100% |

**E**

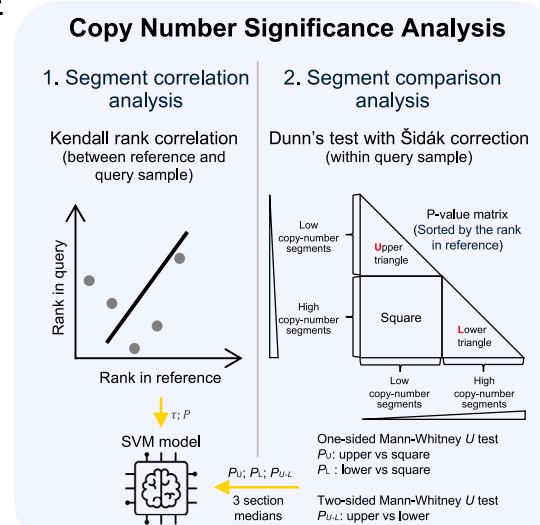

**Copy Number Significance Analysis**

1. Segment correlation analysis

Kendall rank correlation
(between reference and query sample)

Rank in query / Rank in reference

$\tau$; $P$

SVM model

2. Segment comparison analysis

Dunn's test with Šidák correction
(within query sample)

Low copy-number segments — **U**pper triangle
High copy-number segments — Square — **L**ower triangle

P-value matrix (Sorted by the rank in reference)

Low copy-number segments / High copy-number segments

One-sided Mann–Whitney $U$ test
$P_U$: upper vs square
$P_L$: lower vs square

Two-sided Mann–Whitney $U$ test
$P_{U-L}$: upper vs lower

$P_U$; $P_L$; $P_{U-L}$
3 section medians

**F**

0.1% (AUC=0.62)
0.2% (AUC=0.65)
0.3% (AUC=0.74)
0.5% (AUC=0.84)
1.0% (AUC=0.95)
2.0% (AUC=0.97)
3.0% (AUC=0.97)
5.0% (AUC=0.97)
10.0% (AUC=0.97)

◀ **Figure 2. Personalized CNA analysis enables sensitive ctDNA detection and accurate tumor fraction estimation.**

(A) Workflow for Copy Number Similarity Analysis module. informCNA quantifies the extent of tumor DNA shared between reference and query samples by comparing similarity between genome-wide copy number profiles. The resulting Similarity Score is computed using weighted least squares algorithm with LASSO regularization by comparing coverage in each bin. The TF in the query sample is estimated by multiplying the Similarity Score by the TF of the reference sample. (B) Boxplot of informCNA estimated TFs in in silico synthetic tumor DNA mixtures, with the central line representing the median, the bounds of box indicating the interquartile range (IQR; 25% and 75% percentiles), and whiskers extend to the most extreme values within 1.5 times IQR. Outliers exceeding the whiskers are plotted individually to represent the full data range, including minimum and maximum values. The DNA admixtures were analyzed by comparing them to matched cancer cell line DNA at equivalent sequencing depth for TF estimation. This includes a total of 180 runs for each dilution level spanning six sequence depth datasets (see text). WGS data with coverage below 0.5x were excluded from the analyses. (C) Receiver operating characteristic (ROC) curves for tumor DNA detection using informCNA estimated TF. In silico random sub-sampling of reads from non-cancer plasma cfDNA at corresponding equivalent depths were used as negative controls. ROC curves for TFs of 2% to 10% are overlapping, all with an area under the curve (AUC) of 1. (D) Sensitivity of informCNA tumor detection across different specificities in the in silico diluted tumor DNA admixtures. It illustrates the trade-offs between sensitivity and specificity depending on TF. (E) Workflow for Copy Number Significance Analysis module. This complementary approach consists of two analyses: (1) segment correlation analysis between reference and query samples and (2) segment comparison analysis, which assesses copy number changes in the query sample using segmentation information from the reference sample. A support vector machine (SVM) model was trained on eight statistical features for ctDNA detection. (F) ROC curves for differentiating in silico synthetic DNA mixture samples with and without tumor DNA only using the Copy Number Significance Analysis module. Source data are available online for this figure.

performed in silico random down-sampling of WGS data from ten different cancer cell lines (Ghandi et al, 2019) across ten sequencing depths (0.1x, 0.2x, 0.3x, 0.4x, 0.5x, 1x, 2x, 3x, 4x and 5x; Table EV1; Methods). As shown in Fig. EV1A, average Similarity Scores remained consistently high (>99%) except at 0.1x depth, which showed a slightly lower average (~98.4%). This demonstrates that the Similarity Score can robustly and accurately detect CNAs even at low coverage, enabling effective copy number analysis without the need for deep sequencing.

Next, we simulated query samples with varying TFs by creating in silico admixtures of tumor-derived WGS reads (Ghandi et al, 2019) with cfDNA WGS data from healthy donors (Wang et al, 2025) (Methods). For each of the ten different cancer cell lines and ten sequencing depths, we generated dilution series spanning 14 tumor fractions (0%, 0.0001%, 0.001%, 0.01%, 0.05%, 0.1%, 0.2%, 0.3%, 0.5%, 1%, 2%, 3%, 5%, and 10%) with three replicates each, resulting in 4200 mixed DNA WGS datasets ("samples"). Replicates at depths $\geq 0.2x$ showed high reproducibility, with average Similarity Score > 99.3% across all TFs (Fig. EV1B).

To validate linearity, we calculated Similarity Score for each DNA admixture (query) by comparing it to matched cancer cell line WGS data at the same sequencing depth (reference) and compared the Similarity Score to known tumor DNA fractions in admixture. As shown in Fig. EV2, Similarity Scores were strongly correlated with the true tumor fraction across all sequencing depths (Pearson's $r \geq 0.99$, $P < 0.001$). These results confirm that the Similarity Score can reliably estimate ctDNA fraction in the query sample.

### ctDNA detection using the tumor-informed CNA Similarity Score

To evaluate the LLOD of the Copy Number Similarity Analysis, we analyzed DNA admixtures containing varying fractions of cancer cell line DNA (or TFs). Estimated TFs were calculated by multiplying each admixture's Similarity Score by the tumor purity of the reference cancer cell line (Ghandi et al, 2019). In silico down-sampled healthy plasma cfDNA samples at equivalent sequencing depths served as normal controls.

As shown in Fig. 2B, DNA admixtures containing 10% down to 0.2% tumor DNA yielded median TF estimates above zero,

indicating that informCNA enables ctDNA detection at a LLOD of 0.2% TF. We further assessed the ability of informCNA to classify samples as containing or not containing ctDNA based on the estimated TF in each admixture. At a cell-line DNA concentration of 0.2%, the Similarity Score achieved an area under the receiver operating characteristic (ROC) curve (AUC) of 0.67 (Figs. 2C and EV3), with a sensitivity of 54% at a specificity of 80% (Fig. 2D). Notably, the Similarity Score attained 100% sensitivity and 100% specificity at TFs of 2% or higher (AUC = 1; Fig. 2C,D). Cancer types with high genomic instability exhibited improved detection performance (Fig. EV4). These results demonstrate the superior sensitivity and specificity of the Similarity Score for ctDNA detection compared with tumor-naive analyses.

### Complementary machine learning model assists ctDNA detection

The estimation of TF in reference samples can sometimes be biased by different fitting algorithms used to jointly model TF and ploidy (Poell et al, 2019), and it may also be affected by the LLOD of the chosen software, for example, 3% for ichorCNA (Adalsteinsson et al, 2017), 5% for ACE (Poell et al, 2019), and 20% for Rascal (Sauer et al, 2021). To mitigate this issue in informCNA, we developed a complementary module, termed "Copy Number Significance Analysis", enabling ctDNA detection without relying on the TF estimation from reference samples (Fig. 2E, Methods).

We first designed a segment correlation analysis using Kendall rank correlation to compare genome-wide copy number profiles between reference and query samples (Fig. 2E, Methods). Consecutive genomic regions with similar copy number values are defined as segments. We hypothesized that positive correlations between reference and query segments would indicate subtle coverage perturbations caused by shared CNAs from ctDNA, since background cfDNA, primarily of hematopoietic origin, typically displays a flat copy number profile.

Additionally, we developed a segment comparison analysis to detect ctDNA by performing statistical pairwise comparisons across all segments within the query sample (Fig. 2E, Methods). The resulting $p$-values quantify the segment-level copy number differences. We hypothesized that the presence of ctDNA in the query sample would increase the magnitude of these differences,

yielding more significant *p*-values when comparing segments with larger rank differences (Chén et al, 2023). Segment ranks were derived from the reference sample.

We then combined the descriptive statistics and test results from both analyses to construct a support vector machine (SVM)-based machine learning model, incorporating eight selected features derived from the segment correlation and segment comparison analyses (Fig. 2E, Methods). The SVM output is a binary classification for the Copy Number Significance Analysis module. As shown in Fig. 2F, this model demonstrated discriminatory performance comparable to that of the Similarity Score across varying TFs, achieving AUCs of 0.65 and 0.97 at 0.2% and 2% TFs, respectively.

To reduce false positives, we defined ctDNA positivity in informCNA as requiring both (1) a TF ≥ 0.2% estimated by the Similarity Score and (2) a "True" prediction by the SVM-based Copy Number Significance Analysis module.

## Validation of informCNA using ctDNA in vitro dilutions

We next sought to validate the sensitivity of informCNA for ctDNA detection using two in vitro cfDNA dilution series created by mixing cfDNA extracted from plasma samples of two breast cancer patients (containing ctDNA) into cfDNA extracted from plasma samples of two healthy donors. Six different ctDNA fractions were generated with approximate TF of 10.0%, 3.2%, 1.0%, 0.32%, 0.10%, and 0.032% (Methods). The mixed samples and sequencing data were generated in one laboratory, and bioinformatic analysis was performed in a separate laboratory, blinded to the sample classification. As shown in Fig. 3, informCNA robustly detected ctDNA in these dilution series down to a TF of 0.32% for both series. In contrast, ichorCNA detected ctDNA only at TFs above 3.2% in both series (Fig. 3A,B).

## Benchmarking of tumor-naive and tumor-informed approaches

We compared the performance of informCNA with tumor-naive copy number aberration analysis approaches, t-MAD (Mouliere et al, 2018) and ichorCNA (Adalsteinsson et al, 2017), and a tumor-informed approach, MRDetect-CNA (Zviran et al, 2020). We analyzed sWGS sequencing data of cfDNA from post-operative plasma samples collected at serial time points (*n* = 30) from three patients with stage IV breast cancer (Santonja et al, 2023). The "ground truth" ctDNA fraction was estimated using the Integrated Mutant Allele Fraction (IMAF) from the INtegration of VAriant Reads (INVAR) analysis of targeted capture sequencing of paired plasma samples (Santonja et al, 2023; Wan et al, 2020; Mouliere et al, 2018). Briefly, INVAR employs statistical models for error suppression to reduce sequencing errors and alignment artifacts, enabling precise quantification of ctDNA allele fractions (IMAF) by analyzing a large number of SNVs per sample. When applied to hybrid capture sequencing targeting thousands of patient-specific mutations, INVAR detected tumor DNA at allele frequencies as low as 0.00024% (2.4 ppm) (Santonja et al, 2023). As shown in Fig. 4A, the median ctDNA fraction across follow-up plasma samples was 0.25% IMAF, ranging from 0.002% to 13.9%. IMAF measures the fraction of mutant (haploid) alleles and does not account for wild-type alleles in the (generally diploid) tumor genome, whereas copy-

number–based TF quantifies total ctDNA fraction and is approximately twice as high as IMAF.

In t-MAD analysis (Fig. 4B), applying the previously reported TF cutoff of 1.5% (0.015 t-MAD score) (Mouliere et al, 2018), t-MAD identified ctDNA in 10/30 plasma samples, namely 8/14, 2/7, and 0/9 time points for patients P-IV-01, P-IV-02, and P-IV-03, respectively. Notably, all 10 time points with IMAF above 1% (approximately 2% TF) were detected by t-MAD, demonstrating high sensitivity.

Using ichorCNA, 12/30 plasma samples had TFs exceeding the reported LLOD of 3%, including 9/14, 3/7, and 0/9 time points in the three patients, respectively (Fig. 4C). ichorCNA detected all 10 time points above 1% IMAF and identified two additional time points at 0.66% and 0.88% IMAF, showing higher sensitivity than t-MAD.

For tumor-informed approaches, sequencing data of primary or metastatic tumor tissue from patients at a median depth of 3.8x served as tumor-guided references (Santonja et al, 2023). MRDetect-CNA analysis was conducted under suboptimal conditions, as tumor CNA segments were identified using sWGS of tumor DNA without matched buffy coat sample (Methods). MRDetect-CNA detected ctDNA in 14/30 plasma samples (Z-score > 0), including 11/14, 0/7, and 3/9 time points from the three patients, respectively (Fig. 4D). It detected 5/10 time points above 1% IMAF, and 5/8 time points with IMAF between 0.1% and 1%.

As shown in Fig. 4E, informCNA identified ctDNA in 18/30 samples, including 12/14 time points in patient P-IV-01, 5/7 time points in patient P-IV-02 and 1/9 samples in patient P-IV-03. It detected ctDNA in all 18 plasma samples with IMAF ≥ 0.1% (approximately 0.2% TF; LLOD), without any false positives in samples with IMAF < 0.1%, demonstrating the highest sensitivity among the tested assays.

Correlation analyses between TF estimates and IMAF values from INVAR showed high concordance for informCNA (Pearson's $r = 0.95$, $P = 4.05 \times 10^{-16}$; Fig. EV5D), and ichorCNA (Pearson's $r = 0.95$, $P = 1.65 \times 10^{-15}$; Fig. EV5B), followed by t-MAD (Pearson's $r = 0.92$, $P = 1.32 \times 10^{-12}$; Fig. EV5A) and MRDetect-CNA (Pearson's $r = 0.02$, $P = 0.90$; Fig. EV5C). In samples with low TFs (IMAF < 3%), informCNA exhibited the highest correlation (Pearson's $r = 0.98$, $P = 1.32 \times 10^{-17}$), outperforming ichorCNA (Pearson's $r = 0.94$, $P = 4.25 \times 10^{-12}$), t-MAD (Pearson's $r = 0.90$, $P = 3.34 \times 10^{-9}$) and MRDetect-CNA (Pearson's $r = 0.33$, $P = 0.11$), indicating superior accuracy in quantifying ctDNA in low TF samples (Fig. 4F).

Next, we evaluated the specificity using sWGS plasma samples (*n* = 70) from healthy donors (Mouliere et al, 2018). For tumor-informed assays, each sample was compared with the breast cancer tissue references (*n* = 3). Tumor-informed approaches showed lower false positive rates (Dataset EV1), with informCNA exhibiting the lowest at 0.48% (Fig. 4J), followed by MRDetect-CNA at 0.95% (Fig. 4I). The tumor-naive methods had higher false positive rates, with t-MAD at 4.29% (Fig. 4G) and ichorCNA at 14.29% (Fig. 4H), underscoring the high specificity of informCNA.

## informCNA enables earlier detection of recurrence

The serum CA-125 test is routinely used as a clinical biomarker to detect recurrence in patients with ovarian cancer (Paracchini et al, 2021). An elevation in CA-125 levels (>35 U/mL) is currently

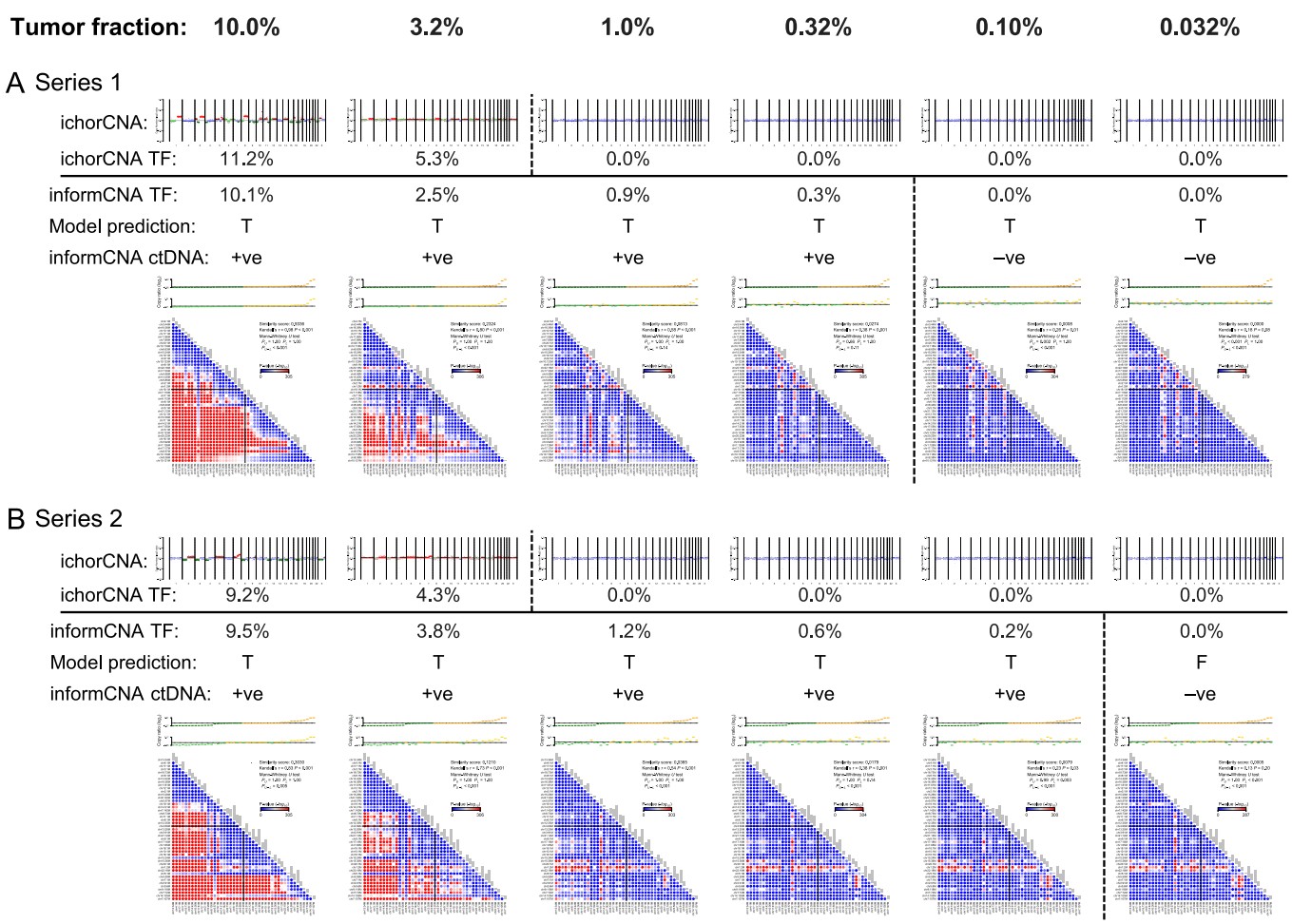

| Tumor fraction: | 10.0% | 3.2% | 1.0% | 0.32% | 0.10% | 0.032% |
|---|---|---|---|---|---|---|
| **A** Series 1 | | | | | | |
| ichorCNA: | | | | | | |
| ichorCNA TF: | 11.2% | 5.3% | 0.0% | 0.0% | 0.0% | 0.0% |
| informCNA TF: | 10.1% | 2.5% | 0.9% | 0.3% | 0.0% | 0.0% |
| Model prediction: | T | T | T | T | T | T |
| informCNA ctDNA: | +ve | +ve | +ve | +ve | −ve | −ve |
| **B** Series 2 | | | | | | |
| ichorCNA: | | | | | | |
| ichorCNA TF: | 9.2% | 4.3% | 0.0% | 0.0% | 0.0% | 0.0% |
| informCNA TF: | 9.5% | 3.8% | 1.2% | 0.6% | 0.2% | 0.0% |
| Model prediction: | T | T | T | T | T | F |
| informCNA ctDNA: | +ve | +ve | +ve | +ve | +ve | −ve |

**Figure 3. ctDNA assessment using in vitro dilution series.**

For Series 1 (**A**) and Series 2 (**B**), plasma cfDNA from two different breast cancer patients was independently diluted with normal plasma from two different healthy donors. This generated ctDNA gradients ranging from 10.0% to 0.032%. The first two rows show copy number profiles and TF estimates derived from ichorCNA. The rest rows display results from informCNA: TF estimated using Copy Number Similarity Analysis (row 3), and ctDNA detection using Copy Number Significance Analysis (row 4), where True (T) indicates tumor-specific signal detected by SVM model and False (F) indicates no tumor-specific signal detected by SVM model. The fifth row summarizes ctDNA status as positive (+ ve) or negative (−ve), with a +ve ctDNA call requiring both TF estimate above the LLOD (0.2%) and a True prediction from the SVM model. A sample was labeled ctDNA +ve only if the estimated ctDNA fraction was ≥0.2% and the SVM model predicted T. The sixth row shows detailed graphical outputs of the Copy Number Significance Analysis by informCNA. The top panel displays the segment correlation analysis, comparing copy number ratios of each segment between the reference (upper line) and query samples (lower line). Segments are ranked by copy number in the reference sample (from lowest to highest, left to right), with the left (lower) half colored green and right (higher) half colored yellow. The bottom panel shows a p-value matrix in the segment comparison analysis, aligned with the segment order above (from lowest to highest, left to right and top to bottom). In the lower-triangular p-values matrix, each dot represents a Dunn's test result, with colors transitioning from blue to red to indicate increasing level of statistical significance. The prominent red region in the lower-left corner corresponds to comparisons between the lowest to highest ranked segments, with the largest differences. The x- and y-axes are labeled with start coordinate of each segment. Gray bars along the diagonal indicate segment lengths. Source data are available online for this figure.

considered the gold standard for indicating potential disease progression (Gupta and Lis, 2009). However, its sensitivity is limited. To evaluate the potential of informCNA for earlier recurrence detection compared to the protein-based CA-125 test, we tested informCNA on 177 sWGS data of 18 patients with high-grade serous epithelial ovarian cancer, as described in a previously published study (Paracchini et al, 2021). These patients underwent platinum-based chemotherapy and had both baseline tumor DNA and longitudinal follow-up plasma cfDNA sequenced to ~1x coverage. Recurrence was assessed by radiological imaging and serum CA-125 test (Paracchini et al, 2021).

For each patient, multi-regional tumor biopsies were available, with tumor purity estimated using ichorCNA (Paracchini et al, 2021). The highest-purity sample (median TF: 58.4%; range: 7.8% to 83.2%) was used as a reference for sensitive ctDNA detection in subsequent plasma samples. Cancer recurrence was identified as the first time point at which plasma cfDNA showed both a positive ctDNA signal identified by informCNA and an increased TF compared to the preceding time point. As shown in Fig. 5A, informCNA detected recurrence earlier than the CA-125 test in 11 patients, both methods detected recurrence at the same time in 3 patients and neither detected relapse in 4 patients. Overall,

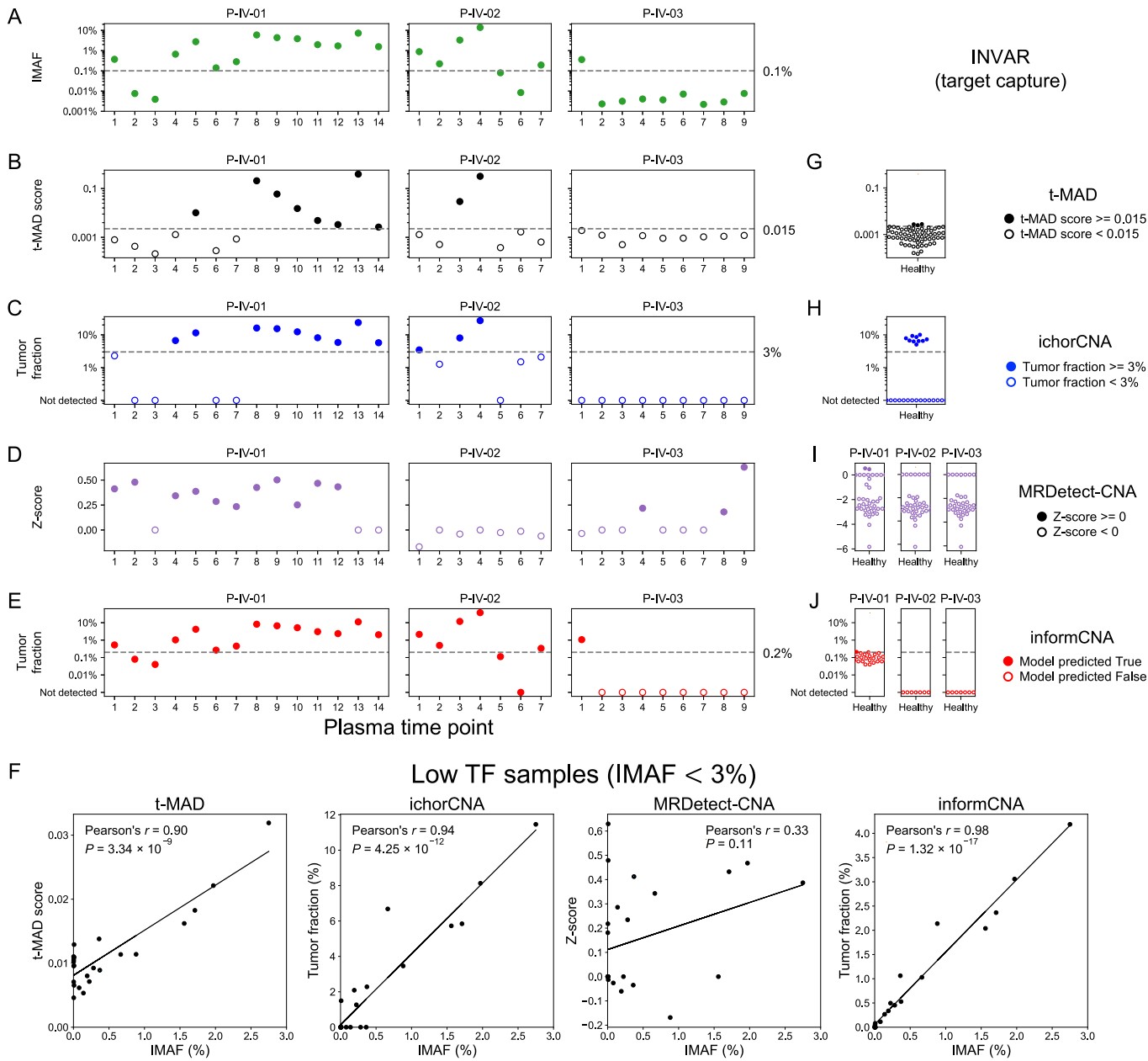

**Figure 4. ctDNA detection and tumor fraction correlation analysis across different assays.**

(A) ctDNA fractions are represented as Integrated Mutant Allele Fraction (IMAF) from INtegration of VAriant Reads (INVAR) analysis using target-capture sequencing of plasma cfDNA. All other figures are based on sWGS of plasma cfDNA. In the tumor-naive setting, (B) t-MAD scores and (C) TF estimated by ichorCNA were used to detect ctDNA. In the tumor-informed setting, guided by sWGS of tumor tissue, (D) Z-score generated by MRDetect-CNA and (E) TF estimated by informCNA were employed to detect ctDNA. (F) Correlation between IMAF values and the output values of t-MAD, ichorCNA, MRDetect-CNA, and informCNA in low TF cfDNA samples with IMAF < 3%. Panels on the right show boxplots of ctDNA levels in plasma cfDNA samples from healthy individuals estimated by t-MAD (G), ichorCNA (H), MRDetect-CNA (I, informed analysis calculated separately informed by each of the 3 patients), and informCNA (J, informed analysis calculated separately informed by each of the 3 patients). Source data are available online for this figure.

informCNA detected relapse at a median of 231 days earlier than radiological imaging (range: 0 to 569 days) and a median of 112 days earlier than CA-125 test, with a range from 0 to 366 days ($P = 0.01$, Wilcoxon signed-rank test; Fig. 5B).

We also observed a higher consistency between TFs estimated by informCNA and CA-125 levels, whereas TFs estimated by

ichorCNA showed divergent trends at several time points in a few patients (Fig. EV6). Additionally, manual curation of ichorCNA results (Paracchini et al, 2021) revealed discrepancies compared to the default output (Fig. EV6). These differences may confound clinicians and are sometimes influenced by stochastic noise and individual interpretation. The results suggest that

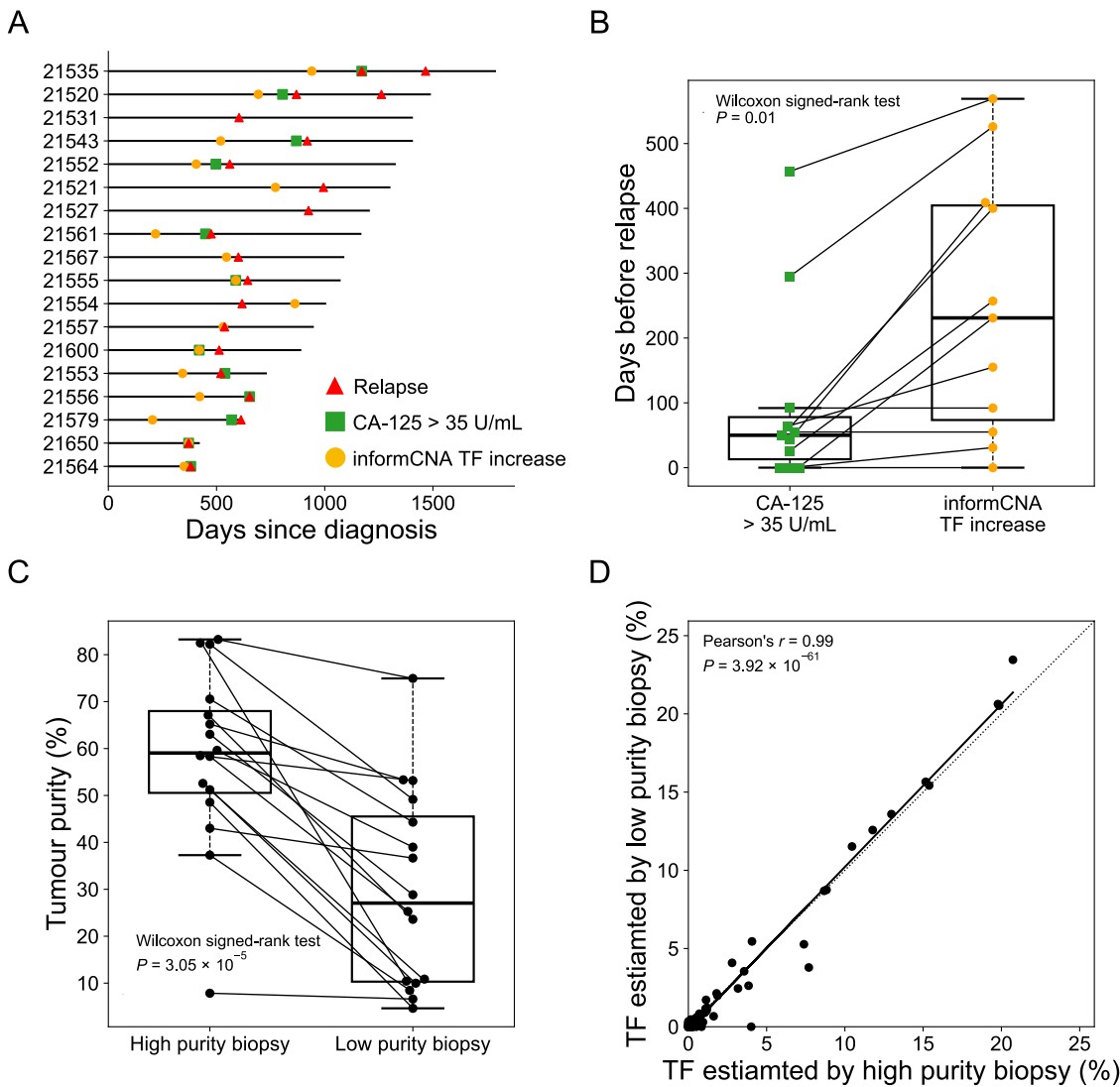

**Figure 5. informCNA performance in longitudinal monitoring of ovarian cancer patients.**

(A) Swimmer plot comparing the timing of recurrence detected by informCNA (yellow circles), CA-125 testing (green squares), and radiology imaging (red triangles). Patient IDs ($n = 18$) are shown on the y-axis. (B) Boxplot showing the number of days by which CA-125 and informCNA detected relapse earlier than radiologic imaging in 18 ovarian cancer patients. The central line represents the median, the box denotes the interquartile range (IQR; 25% to 75% percentiles), and the whiskers extend to the most extreme values within 1.5 times IQR, with all individual data points shown. (C) Tumor purity comparison between the high- and low-purity biopsies collected from different tumor regions of the same patient ($n = 16$), with central line, box, and whiskers of boxplot defined as described above. (D) Correlation between TFs estimated by informCNA using either the high- or low-purity biopsy as the reference sample in the same patient. Source data are available online for this figure.

informCNA is more robust than tumor-naive CNA analysis methods and demonstrates better concordance with the CA-125 test.

## informCNA shows consistency against tumor heterogeneity

Tumor spatial heterogeneity, combined with sampling bias, may cause variability in tumor purity and CNA profiles across tissue biopsies from the same patient, resulting in uncertainty in reference CNA profiling. To assess whether the selection of reference samples influences TF estimation in query samples, informCNA was further evaluated with respect to tumor heterogeneity using multi-regional tumor biopsies, including tumor tissues from the ovary, omentum, and peritoneum (Paracchini et al, 2021).

As shown in Fig. 5C, tumor samples with high purity (median TF: 59.0%; range: 7.8% to 83.2%) had 2.2 times higher TF ($P = 3.05 \times 10^{-5}$, Wilcoxon signed-rank test) than the low purity tumor biopsies (median TF: 27.1%; range: 4.6% to 74.9%) from the same patient. informCNA analysis was performed on the same query plasma sample using both high- and low-purity tumor biopsies as references, compared TF estimates produced by these

two different references. Intriguingly, informCNA demonstrated robust ctDNA fraction estimations regardless of the purity of the reference samples (Pearson's $r = 0.99$, $P = 3.92 \times 10^{-61}$; Figs. 5D and EV7).

## informCNA ctDNA monitoring in plasma-guided model

In many clinical situations, tumor tissue biopsy is not available. To address this, we explored the use of informCNA to monitor ctDNA dynamics through a plasma-guided model, without relying on tumor biopsy (Fig. 6A). We analyzed 25 sWGS plasma DNA samples from five melanoma patients in our previously published study (Wan et al, 2020). INVAR analysis, which integrates tumor-identified mutations from target-capture sequencing of paired plasma samples, was used to quantify ctDNA IMAF (Wan et al, 2020). Five baseline plasma samples collected before treatment (median IMAF:28.9%; range: 11.0% to 38.0%), served as references for tumor CNA detection and guided ctDNA detection in the 20 follow-up plasma samples (median IMAF: 8.5%; range: 0.003% to 64.0%).

As shown in Fig. 6B, TFs estimated by the plasma-guided model displayed a strong correlation with IMAF in the paired plasma samples (Pearson's $r = 0.91$, $P = 7.07 \times 10^{-10}$). Notably, informCNA detected ctDNA in all 14 plasma samples with IMAF ≥ 0.1%, as well as one additional sample at the third time point of Patient 62 with an IMAF of 0.05% (Fig. 6C). informCNA also correctly classified the remaining five ctDNA-negative samples, all of which had IMAF below 0.1% (Fig. 6C). These results demonstrate the feasibility and sensitivity of informCNA for ctDNA detection in a plasma-only context.

## Discussion

In this study, informCNA is introduced as a tumor-informed cfDNA copy number analysis pipeline designed to improve the sensitive, longitudinal monitoring of ctDNA burden in cancer patients using low-coverage WGS. By leveraging prior information of patient-specific, tumor-associated CNAs, informCNA achieves a LLOD of 0.2% ctDNA fraction (TF) with 54% sensitivity at 80% specificity (equivalent to 1000 ppm), which is 15-fold more sensitive than that of state-of-the-art tumor-naive CNA analysis pipeline, ichorCNA (Adalsteinsson et al, 2017). Additionally, informCNA achieves 100% sensitivity and 100% specificity at a TF of 2%, surpassing the 3% LLOD of ichorCNA, which reaches only 95% sensitivity and 91% specificity at that TF level. In a prognostic setting, the lower detection limit of informCNA can facilitate earlier recurrence detection and support timely clinical interventions, without substantially increasing sequencing cost.

Direct comparison of cfDNA CNA analysis pipelines revealed that informCNA demonstrated the lowest false positive rate when applied to plasma samples from healthy individuals. Overall, both tumor-informed CNA analysis assays exhibited higher specificity than tumor-naive approaches, highlighting the advantage of a tumor-informed strategy for ctDNA detection. For the breast cancer cohort, informCNA also showed potential for detecting ctDNA in follow-up plasma samples with IMAF below 0.01%, though it tended to overestimate TF in such cases. This may be explained by patients with higher levels of genomic instability,

where an increased number of CNA segments with greater amplitude can facilitate cancer detection (Andor et al, 2017). Therefore, informCNA might be capable of detecting ctDNA below the nominal 0.2% TF LLOD with increased sensitivity in patients with an extensive burden of CNAs. The diverse CNA landscapes across cancer types also underscore the need for further parameter tuning to optimize sensitivity and specificity for specific cancer types, such as using a bin size that aligns with CNA segment length for optimal copy number profiling.

In prognostic stratification, a single deep-sequencing–based assay for postoperative ctDNA detection, though expensive, can accurately identify MRD and enable precise risk stratification (Zviran et al, 2020; Widman et al, 2024; Black et al, 2025b). However, a cost-efficient, rapid-turnaround assay is essential for frequent monitoring of treatment response, early detection of recurrence, and real-time guidance of therapeutic decisions. informCNA, which is based solely on sWGS data, is designed to address this clinical need. Its low sequencing cost, simple workflow, and fast processing time make it particularly suitable for longitudinal monitoring, where frequent tests may be required throughout treatment. While repeated testing inevitably adds operational costs (e.g., blood collection and staff time), the sequencing and analysis cost per informCNA test is significantly lower than targeted panels or deep WGS, due to shallow coverage and a streamlined workflow. As a result, even with frequent sampling, sWGS remains a cost-effective strategy for real-time monitoring. Importantly, these approaches serve distinct but complementary clinical roles: single high-cost assays are optimal for initial MRD detection and risk stratification, whereas sWGS-based assays provide a practical solution for repeated, rapid assessment during active surveillance or adjuvant therapy.

Compared to tumor-naive methods, tumor-informed approaches require an additional prior biopsy from previous time points or sources, making them especially suitable for prognostic ctDNA monitoring in clinical scenarios where longitudinal collection of multiple biopsy samples is feasible. However, these methods may be confounded by tumor heterogeneity and clonal evolution induced during treatment, potentially leading to variability in CNA profiles derived from reference biopsies and impairing ctDNA detection from subsequent query biopsies. In our validation analyses, informCNA demonstrated the ability to address these challenges, yielding highly concordant results across multi-regional tumor biopsies with substantially different tumor purities. The use of personalized CNAs in a tumor-informed setting also enables more specific and robust quantification of ctDNA fractions, reducing susceptibility to sequencing errors and contamination. Furthermore, informCNA is applicable to a plasma-guided model, accurately estimating ctDNA burden using only longitudinal plasma samples. Leveraging reference liquid biopsy may provide a more comprehensive view of tumor heterogeneity than tissue biopsy, potentially further enhancing the accuracy of ctDNA detection during follow-up cfDNA analysis.

In the prognostic monitoring of cancer patients, routinely used clinical follow-up methods, such as radiological imaging and serum protein tests, have limited sensitivity and specificity. Our results indicate that informCNA has the potential to be more sensitive than conventional monitoring approaches, enabling anticipation of disease progression several months earlier than radiological

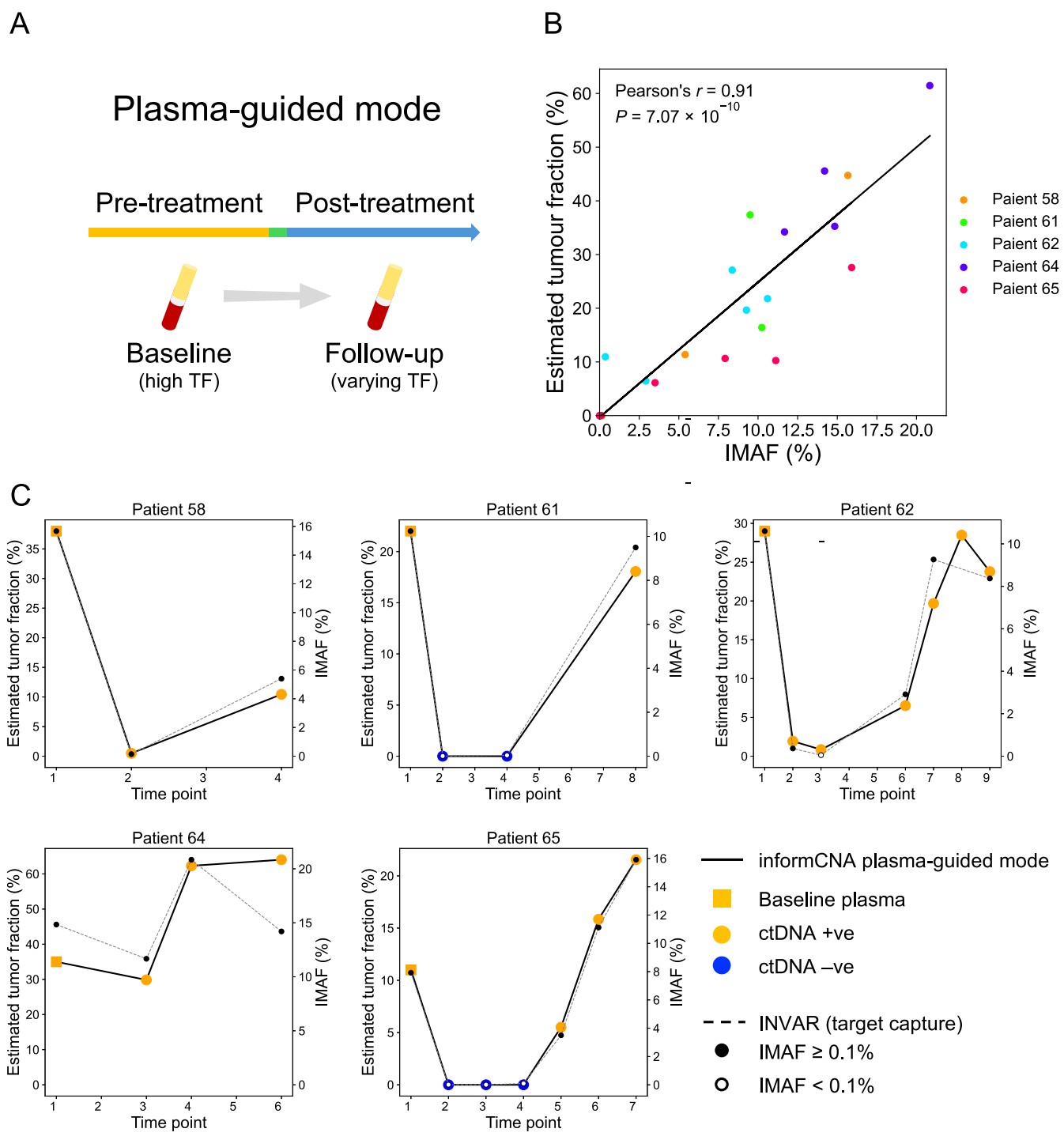

Figure 6. Monitoring ctDNA dynamics using only liquid biopsies.

(A) Principle of plasma-guided model of informCNA for ctDNA detection. (B) Correlation between IMAF values from paired target-capture plasma samples and TF estimated by informCNA in sWGS of plasma cfDNA. Five baseline and twenty follow-up plasma samples were collected from five melanoma patients. The tumor fractions of baseline plasma samples were estimated using ACE (Poell et al, 2019). (C) Comparison of IMAF and TFs estimated by informCNA in the plasma-guided model. Solid lines represent informCNA results. Baseline plasma samples are shown as yellow squares, and follow-up plasma samples are represented as circles. Yellow circles represent ctDNA +ve time points, while blue circles represent ctDNA −ve ones. Dashed lines represent IMAF values obtained from target-capture sequencing of paired plasma samples. Filled circles indicate time points where IMAF ≥ 0.1%, corresponding to the 0.2% LLOD of informCNA, while open circles represent IMAF values below this threshold. Source data are available online for this figure.

imaging or CA-125 analysis. Tracking prognostic ctDNA dynamics with informCNA may help assess treatment response, allow for earlier recurrence detection, guide therapeutic decisions, and ultimately improve patient outcomes. Further studies are needed to clarify its application in clinical workflows and to evaluate performance in larger patient cohorts.

This study also has limitations. For instance, informCNA was evaluated only in patients with advanced cancer stages. In early-stage tumors, tissue or plasma samples collected before treatment may contain very low fractions of tumor DNA, creating challenges for detecting tumor-associated informative CNAs, especially in the plasma-guided model. Furthermore, clonal hematopoiesis associated CNAs may confound analyses in the plasma-guided mode, as persistent non-tumor CNAs in plasma cfDNA may interfere with Copy Number Similarity Analysis and impair informCNA performance, potentially resulting in overestimation of TF due to false-positive signals (Saiki et al, 2021). Occasionally, tumor biopsies are formalin-fixed and paraffin-embedded (FFPE). Although FFPE samples generally show high concordance with fresh-frozen tumor tissues, they can be limited by low tumor cellularity and copy-number artifacts (Basyuni et al, 2024). Strategically, informCNA does not directly calculate the TF of a query sample but instead provides a Similarity Score relative to the TF of reference samples. Consequently, accurate TF estimation requires sufficient tumor DNA in the reference biopsies, ensuring that CNA segments are detectable in at least one tumor-naive assay. These limitations warrant further investigation in real-world clinical settings.

In summary, this study demonstrates the feasibility of using informCNA on sWGS plasma cfDNA to sensitively and accurately detect ctDNA and monitor disease progression in cancer patients in a cost-effective manner.

## Methods

**Reagents and tools table**

| Reagent/Resource | Reference or Source | Identifier or Catalog Number |
|---|---|---|
| **Experimental models** | | |
| NA | NA | NA |
| **Recombinant DNA** | | |
| NA | NA | NA |
| **Antibodies** | | |
| NA | NA | NA |
| **Oligonucleotides and other sequence-based reagents** | | |
| NA | NA | NA |
| **Chemicals, Enzymes and other reagents** | | |
| NA | NA | NA |
| **Software** | | |
| BWA | https://github.com/lh3/bwa | Version 0.7.18 |
| Picard | https://github.com/broadinstitute/picard | Version 3.2.0 |
| Samtools | https://github.com/samtools/samtools | Version 1.20 |
| QDNAseq | https://github.com/ccagc/QDNAseq | Version 1.41.3 |
| GenMap | https://github.com/cpockrandt/genmap Pockrandt et al, 2020 | Version 1.3.0 |
| ichorCNA | https://github.com/broadinstitute/ichorCNA Adalsteinsson et al, 2017 | Version 0.2.0 |
| ACE | https://github.com/tgac-vumc/ACE Poell et al, 2019 | |
| t-MAD | https://github.com/sdchandra/tMAD Mouliere et al, 2018 | |
| MRDetect-CNA | Zviran et al, 2020 | |
| CNVkit | https://github.com/etal/cnvkit Talevich et al, 2016 | Version 0.9.11 |
| Python | https://python.org | Version 3.12 |
| pysam | https://pysam.readthedocs.io/en/latest/api.html | Version 0.22.1 |
| pyBigWig | https://github.com/deeptools/pyBigWig | Version 0.3.23 |
| cvxpy | https://github.com/cvxpy/cvxpy/ | Version 1.6.1 |
| matplotlib | https://github.com/matplotlib/matplotlib | Version 3.10 |
| numpy | https://github.com/numpy/numpy | Version 2.2.0 |
| pandas | https://github.com/pandas-dev/pandas | Version 2.2.2 |
| ruptures | https://centre-borelli.github.io/ruptures-docs/ | Version 1.1.9 |
| scipy | https://github.com/scipy/scipy | Version 1.15.1 |
| scikit_posthocs | https://github.com/maximtrp/scikit-posthocs | Version 0.11.2 |
| scikit-learn | https://github.com/scikit-learn/scikit-learn | Version 1.5.2 |
| informCNA | https://github.com/nrlab-CRUK/informCNA This study | |
| **Other** | | |
| KAPA HyperPrep Kit | Roche | |
| Illumina NovaSeq 6000 | Illumina | |

## Study design

We acquired paired-end sequencing data from a previous study (Paracchini et al, 2021), which included sWGS of 185 plasma samples and 109 matched tumor biopsies from 46 patients with high-grade serous epithelial ovarian cancer, from the European Genome-Phenome Archive (EGA; accession number EGAD00001006422). We selected 18 patients who had baseline tumor biopsies with detectable CNAs and 177 serial follow-up plasma samples available.

For the in silico dilution series, we obtained 10 cancer cell lines (median coverage: 28.9x; range: 26.5 to 33.8x; Table EV1) from the Cancer Cell Line Encyclopedia (Ghandi et al, 2019), as well as a pool of healthy plasma samples (coverage: 156.6x) from ten healthy donors (Wang et al, 2025). Additionally, paired-end sequencing data were obtained from previous studies (Mouliere et al, 2018; Wan et al, 2020; Santonja et al, 2023), including tissue samples of three patients with breast cancer (median coverage: 3.8x; range: 3.7 to 3.9x), and a total of 137 plasma samples. The plasma DNA samples included 70 healthy plasma samples (median coverage: 0.6x; range: 0.4 to 1.0x), 30 postoperative plasma samples (median coverage: 0.9x; range: 0.5 to 1.3x) from the three aforementioned breast cancer patients, and 25 plasma DNA samples from five patients with melanoma (median coverage: 0.4x; range: 0.2 to 1.0x).

Furthermore, we generated 12 plasma DNA admixtures for an in vitro dilution series (median coverage: 4.3x; range: 3.5 to 4.9x). The study protocol was conducted as part of the Advanced Breast Cancer Biopsy (ABC-Bio) trial (CCR3991, REC ID: 14/LO/0292) and was approved by the NHS Health Research Authority, Research Ethics Committee London-Chelsea. Written informed consent was obtained from each patient in accordance with regulatory requirements, good clinical practice, the World Medical Association (WMA) Declaration of Helsinki, and the Department of Health and Human Services Belmont Report. Patients consented to either a biopsy of metastatic disease or access to an archival biopsy of recurrent disease (Pearson et al, 2020).

## In vitro dilution of plasma cfDNA

Two series of in vitro dilution samples were generated using plasma cfDNA samples from two different breast cancer patients, with plasma cfDNA from two healthy donors serving as the diluent. DNA libraries were constructed using the KAPA HyperPrep Kit (Roche) using UDI Illumina barcodes. Series 1 began at 10.85% TF, while Series 2 started at 31.5% TF, which was initially diluted to approximately 10%. TFs were estimated by ichorCNA in default setting. Both Series 1 and Series 2 underwent half-log serial dilutions at a 3.16:1 ratio using plasma cfDNA from healthy donors. The in vitro plasma cfDNA mixture samples were sequenced using the Illumina NovaSeq S2 200-cycle flow cell.

## Analyses of sequencing data

Paired-end DNA sequencing reads were aligned to the human reference genome hg38 using BWA-MEM (v0.7.18). PCR duplicates were identified and removed using MarkDuplicates (Picard v3.2.0). The aligned reads were sorted and indexed with SAMtools (v1.20; sort and index commands). Sequencing coverage was calculated using CollectWgsMetrics (Picard v3.2.0). Tumor purity in tissue and plasma DNA was estimated using ACE (Poell et al, 2019), which depends on the QDNAseq (v1.41.3) R package. In silico dilution of cancer cell line DNA with cfDNA from healthy individual was performed by down-sampling BAM files to specific numbers of paired-end reads using DownsampleSam (Picard v3.2.0), followed by merging the tumor and normal DNA fragments into a combined BAM file using SAMtools (v1.20; merge command).

## informCNA: ctDNA fraction estimation using Copy Number Similarity Analysis

The mappability of the human reference genome (hg38) was assessed using GenMap (Pockrandt et al, 2020). Low-mappability regions, including centromeres and telomeres, were excluded by applying a cutoff with a mappability score less than 0.2. The remaining high-mappability regions were divided into 100 kb bins (windows) with 10 kb steps.

Aligned reads with mapping quality ≥30 were counted based on overlap with each bin using pysam (v0.22.1) Python module. The coverage of each bin was calculated on a log2 scale of the read counts. The log2 copy ratio in each bin of the autosomal chromosomes was then corrected subsequently for GC content (UCSC Genome Browser), mappability and replication timing (Zhao et al, 2020) using robust spline regression (scipy v1.15.1). Copy number segments, which are consecutive genomic regions with similar copy number status, were identified by segmentation across all the bins using ruptures (v1.1.9) Python module, based on the Pelt algorithm with linear penalty (Truong et al, 2020).

Segmentation was first applied to the reference sample to differentiate consecutive bins exhibiting copy number aberrations. The copy number for each segment was calculated as the median copy number of all bins within that segment. This segmentation was then projected onto the query sample to calculate copy number values of corresponding regions in the same manner.

We employed a weighted least squares (WLS) algorithm with LASSO regularization (cvxpy v1.6) to compute genome-wide similarity between reference and query DNA samples, termed the copy number Similarity Score. The Similarity Score (denoted as $x$) is formulated as:

$$\min_{x \geq 0} \left\{ \sum_{i=1}^{k} w_i \left( a_i^T x - b_i \right)^2 + \lambda \| x \|_1 \right\}$$

where $a_i$ and $b_i$ are copy number values for genomic bin $k$ in reference and query samples, respectively and $\lambda$ is the $\ell 1$ regularization penalty mitigating noise, enabling robust estimation.

informCNA assigns weights to segments to emphasize focal chromosomal CNAs. The weight of each segment ($w_i$) is calculated as:

$$w_i = \begin{cases} 1, & (\textit{single segment chromosome}) \\ 1 + \frac{CN_{ref} * SD_{qry}}{SD_{ref}}, & (\textit{otherwise}) \end{cases}$$

where $CN_{ref}$ represents the absolute copy number of the targeted segment in the reference sample. $SD_{ref}$ and $SD_{qry}$ represent the segment-level copy number standard deviations compared to other segment(s) within the same chromosome in the reference and query sample, respectively. For chromosomes with only one segment, no additional weight was applied. These weights increase emphasis on informative CNA segments with greater amplitude, enhancing tumor CNA signal detection.

## informCNA: ctDNA detection using Copy Number Significance Analysis

Copy Number Significance Analysis employs an SVM model to determine the presence of ctDNA in a query sample based on two

analyses: segment correlation analysis between the reference and query samples, and segment comparison performed solely on the query sample.

In the segment correlation analysis, the copy number of each segment was calculated as the median of all bins within that segment. Both the reference and query samples used identical segments derived from the reference sample. The correlation between reference and query DNA samples was then calculated using the non-parametric Kendall rank correlation (scipy v1.15.1), based on the ranks of copy number values across all segments.

For the segment comparison analysis of query sample, segmentation was similarly based on the reference sample. Pairwise multiple comparisons were performed between every pair of the segments in the query sample using the bins within those segments, producing a matrix of $p$-values. This comparison used Dunn's test with Sidak $p$-value adjustments, implemented using scikit-posthocs (v0.7.0) Python module.

The $p$-value matrix was organized from low to high copy number order along both axes, based on the segment ranks in the reference sample. Only the lower triangular matrix was shown (Fig. 3), as the upper matrix is symmetric. Segments were categorized into lower and higher copy-number groups relative to the median copy number ranks. The $p$-value matrix was divided into three sections: an upper sub-triangle (comparisons between lower and lower copy-number segments), a lower sub-triangle (comparisons between higher and higher copy-number segments) and a nearly square section (comparisons between lower and higher copy-number segments).

We performed one-sided Mann–Whitney $U$ test (scipy v1.15.1) comparing the $p$-values in the upper sub-triangle section to those in the square section (denoted as $P_U$), and those in lower sub-triangle section to the square section (denoted as $P_L$); as well as a two-sided Mann–Whitney $U$ test comparing $p$-values in the upper and lower sub-triangle sections (denoted as $P_{U-L}$).

The SVM classification model (scikit-learn v1.5.2) integrated eight features from segment correlation and pairwise comparison results. These included Kendall's tau and its associated $p$-value from the correlation analysis; the median $p$-values from the upper, lower sub-triangle, and square sections of the pairwise comparison matrix; and the three comparative $p$-values $P_U$, $P_L$ and $P_{U-L}$.

## informCNA consistency across different bin sizes

In informCNA, genome-wide coverage was represented in 100 kb bins by default. We assessed the effect of bin size on sequencing coverage calculation by testing bins of 50 kb and 200 kb mappable regions. Both reference tumor tissue DNA and query plasma cfDNA samples were analyzed using these bin sizes for comparison. In thirty serial plasma samples from three patients with breast cancer (Santonja et al, 2023), the Similarity Score yielded nearly identical results comparing 50 kb bins to the default 100 kb bins (Pearson's $r = 1$, $P = 6.12 \times 10^{-59}$; Fig. EV8A), as well as 200 kb bins to the 100 kb bins (Pearson's $r = 1$, $P = 1.54 \times 10^{-66}$; Fig. EV8B). Additionally, the SVM model produced the same ctDNA prediction in 29/30 time points using 50 kb and 200 kb bins compared to 100 kb bin (Fig. EV8). These results demonstrate that informCNA is robust across different genomic bin sizes.

## Enhanced informCNA TF estimation using in silico size selection

Selective analysis of short plasma cfDNA fragments (< 150 bp) has been shown to enhance ctDNA detection by enriching tumor-derived DNA fractions (Mouliere et al, 2018). To evaluate this enhancement in informCNA performance, we applied in silico size selection to plasma samples ($n = 30$) from three patients with breast cancer (Santonja et al, 2023). We compared TF estimates derived from overall plasma cfDNA without size selection and from plasma cfDNA restricted to short DNA molecules (< 150 bp). As shown in Fig. EV9A, informCNA detected ctDNA in 18/30 samples, including 12/14 time points in patient P-IV-01, 7/7 time points in patient P-IV-02, and 1/9 time points in patient P-IV-03. Notably, in patient P-IV-02, informCNA identified ctDNA at two additional time points compared to analysis using overall plasma DNA (Fig. 4E), without introducing any false negatives. Further comparison of TF estimates with and without the size selection revealed a strong correlation with a slope of 1.55 (Fig. EV9B), indicating an average 55% increase in ctDNA fraction following in silico size selection.

## Benchmarking of tumor-naive and tumor-informed CNA analyses

t-MAD (https://github.com/sdchandra/tMAD) was performed under default settings with dependence of CNAclinic (https://github.com/sdchandra/CNAclinic) R package and QDNAseq (v1.41.3) R package. Parameters were set for human reference genome hg38.

ichorCNA (v0.2.0) was performed in bin size of 500 kb with default parameters and files, except that the initial normal

---

**The paper explained**

**Problem**

Cell-free DNA (cfDNA) offers great promise as a minimally invasive tool for monitoring treatment response and disease progression in prognostic settings. Previous tumor-informed circulating tumor DNA (ctDNA) detection assays generally provide high sensitivity and specificity for tracking cancer dynamics but require customized panel design, making them costly. In contrast, copy number aberration (CNA) analysis using shallow whole-genome sequencing (sWGS) in a tumor-naive setting is more cost-efficient, but has limited sensitivity.

**Results**

We developed informCNA, a tumor-informed CNA analysis method for detecting ctDNA and quantifying tumor fraction (TF) using sWGS of cfDNA. By leveraging an additional high-TF tumor or liquid biopsy sample, informCNA can detect ctDNA at TF as low as 0.2% in follow-up plasma cfDNA samples. We compared informCNA with three other CNA-based ctDNA detection methods and validated its superior sensitivity and specificity.

**Impact**

informCNA enables cost-effective ctDNA profiling with high sensitivity, allowing accurate tracking of treatment response and earlier detection of disease recurrence. Without substantially increasing sequencing costs, it supports frequent longitudinal monitoring of ctDNA levels. Consequently, informCNA may facilitate more timely treatment decision-making and clinical interventions.

contamination parameter was set to "c(0.5,0.6,0.7,0.8,0.9)" and maximum number of clusters was set to 5.

MRDetect-CNA was performed using all default settings, based on CNAs from tumor tissues identified using CNVkit (v0.9.11) with default parameters. Since only sWGS data of tumor tissues were available, the use of CNVkit to identify tumor CNAs may have limited resolution and accuracy, which inevitably impairs the performance of MRDetect-CNA.

The benchmarking analysis emphasizes methodological differences by considering case contexts rather than implying a one-to-one comparison.

## Analysis of ovarian cancer patients

CA-125 levels and manually curated ichorCNA results (with size selection of cfDNA < 150 bp) of all time points were obtained from the published study (Paracchini et al, 2021). Default ichorCNA (v0.2.0) results were generated using the above-mentioned parameters on the size-selected cfDNA. informCNA was run using all default settings on the size-selected plasma cfDNA data.

# Data availability

The source code of informCNA can be found in the Rosenfeld Group GitHub, https://github.com/nrlab-CRUK/informCNA. Sequencing data generated for the in vitro dilution series are publicly available on Zenodo (https://zenodo.org/records/18403944).

The source data of this paper are collected in the following database record: biostudies:S-SCDT-10_1038-S44321-026-00399-4.

# Peer review information

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

## Acknowledgements

We thank the participants who generously donated samples, as well as the clinical teams involved in the clinical trials. We thank the Genomics, Bioinformatics, Histopathology and Biorepository Core Facilities at the Cancer Research UK Cambridge Institute. We thank the Cancer Molecular Diagnostics Laboratory (CMDL) at the University of Cambridge. We are also grateful to Maurizio D'Incalci's group at IRCCS Humanitas Research Hospital, Rozzano, Milan, Italy. We thank Sara Potente, Luca Beltrame, Sergio Marchini, and Solon Karapanagiotis for their valuable input. We are grateful to Dan Landau's group at Weill Cornell Medical College, New York, NY, USA. We acknowledge funding support from the following sources: Cancer Research UK (C507/A27657, C9545/A29580, SEBINT-2024/100003, C1287/A26886, EDDRPG-May24/100002 & C36857/A27548), NIHR Cambridge Biomedical Research Centre (BRC-1215-20014), The Mark Foundation for Cancer Research (RG95043), the Cancer Research UK Cambridge Centre (C9685/A25177), and Addenbrooke's Charitable Trust (ACT 9800). We thank Breast Cancer Now for funding as part of Program Funding to the Breast Cancer Now Toby Robins Research Centre (Grant Number CTR-Q5-Y3). This work was supported by the Cancer Molecular Diagnostics Lab, University of Cambridge who are funded by Cancer Research UK Cambridge Centre [CTRQQR-2021\100012] and the NIHR Cambridge Biomedical Research Centre (NIHR203312). The views expressed are those of the authors and not necessarily those of the NIHR or the Department of Health and Social Care. NR is supported by infrastructure grants within the CRUK City of London Major Centre Awards [C7893/A26233 and CTRQQR-2021\100004]. AR is supported by CRUK-Royal College of Surgeons of England Clinician Scientist Fellowship [C64667/A27958].

## Author contributions

**Ze Zhou**: Conceptualization; Software; Validation; Investigation; Visualization; Methodology; Writing—original draft; Writing—review and editing. **Ros Cutts**: Resources; Data curation; Writing—review and editing. **Sarah Hrebien**: Resources; Data curation; Writing—review and editing. **Christina X Zhang**: Data curation; Investigation; Writing—review and editing. **Isaac Garcia-Murillas**: Investigation; Writing—review and editing. **Woody Z Zhang**: Investigation; Writing—review and editing. **Alexander M Frankell**: Supervision; Writing—review and editing. **Wendy N Cooper**: Data curation; Supervision; Project administration; Writing—review and editing. **Amit Roshan**: Supervision; Writing—review and editing. **Nicholas C Turner**: Supervision; Writing—review and editing. **Tommy Kaplan**: Supervision; Writing—review and editing. **Nitzan Rosenfeld**: Conceptualization; Supervision; Funding acquisition; Project administration; Writing—review and editing. **Hui Zhao**: Conceptualization; Supervision; Investigation; Project administration; Writing—review and editing.

Source data underlying figure panels in this paper may have individual authorship assigned. Where available, figure panel/source data authorship is listed in the following database record: biostudies:S-SCDT-10_1038-S44321-026-00399-4.

## Disclosure and competing interests statement

ZZ, HZ, and NR are co-inventors on a patent application filed by Cancer Research UK based on the data presented in this study, and managed in accordance with institutional policies.

# Expanded View Figures

A

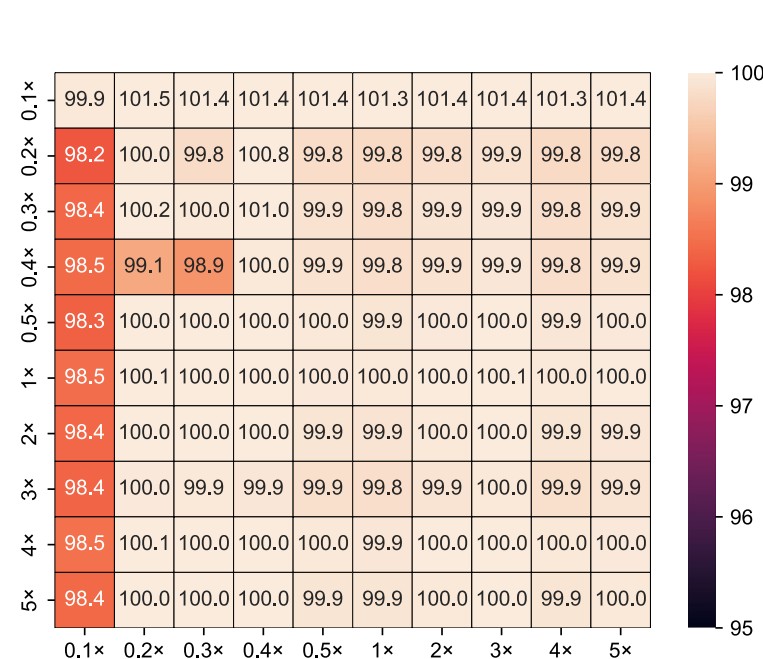

B

**Figure EV1. Robustness and reproducibility assessment of copy number similarity scores in cancer cell line data.**

(A) Average copy number Similarity Scores across different down-sampling depths in sWGS samples from the same cancer cell line (tumor) data for robustness assessment. Similarity Scores exceeding 100% indicate that the query sample is considered to have a higher TF estimate than the reference sample. (B) Copy number Similarity Scores between replicates with the same tumor fraction and sequencing depth in the dilution series for reproducibility assessment, with error bars indicate 95% confidence intervals. In these pairwise comparisons, each replicate, in turn, served as the reference for comparison with the other replicates, resulting in a total of 60 comparisons per condition.

A Overall dilutions B TF ≤ 0.5%

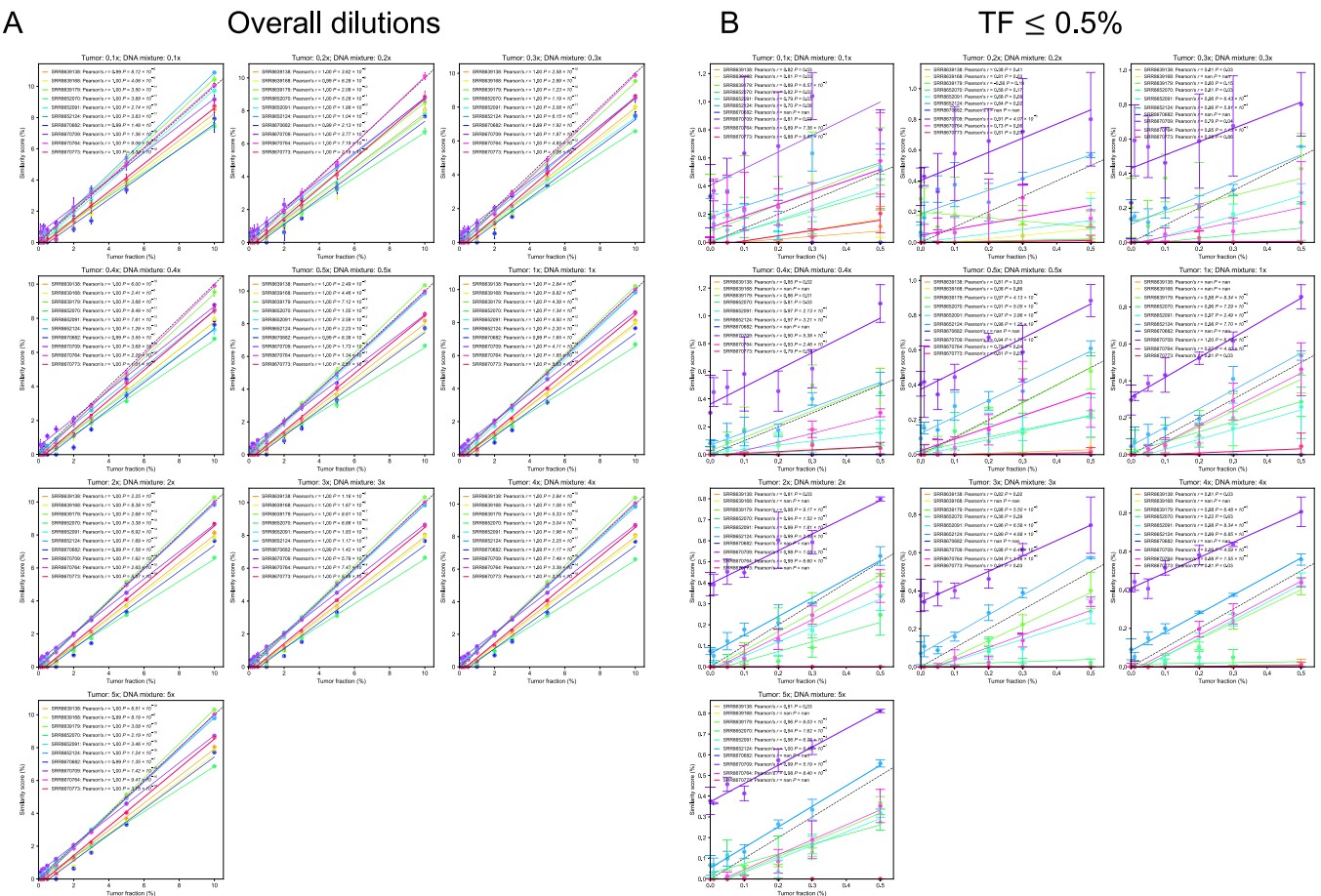

**Figure EV2. Correlations between tumor DNA fractions (TF) and copy number Similarity Scores in diluted cancer cell line samples.**

Error bars show 95% confidence intervals of three replicates for each TF condition. (A) Overall tumor fractions. (B) Samples with tumor DNA fractions ≤ 0.5%.

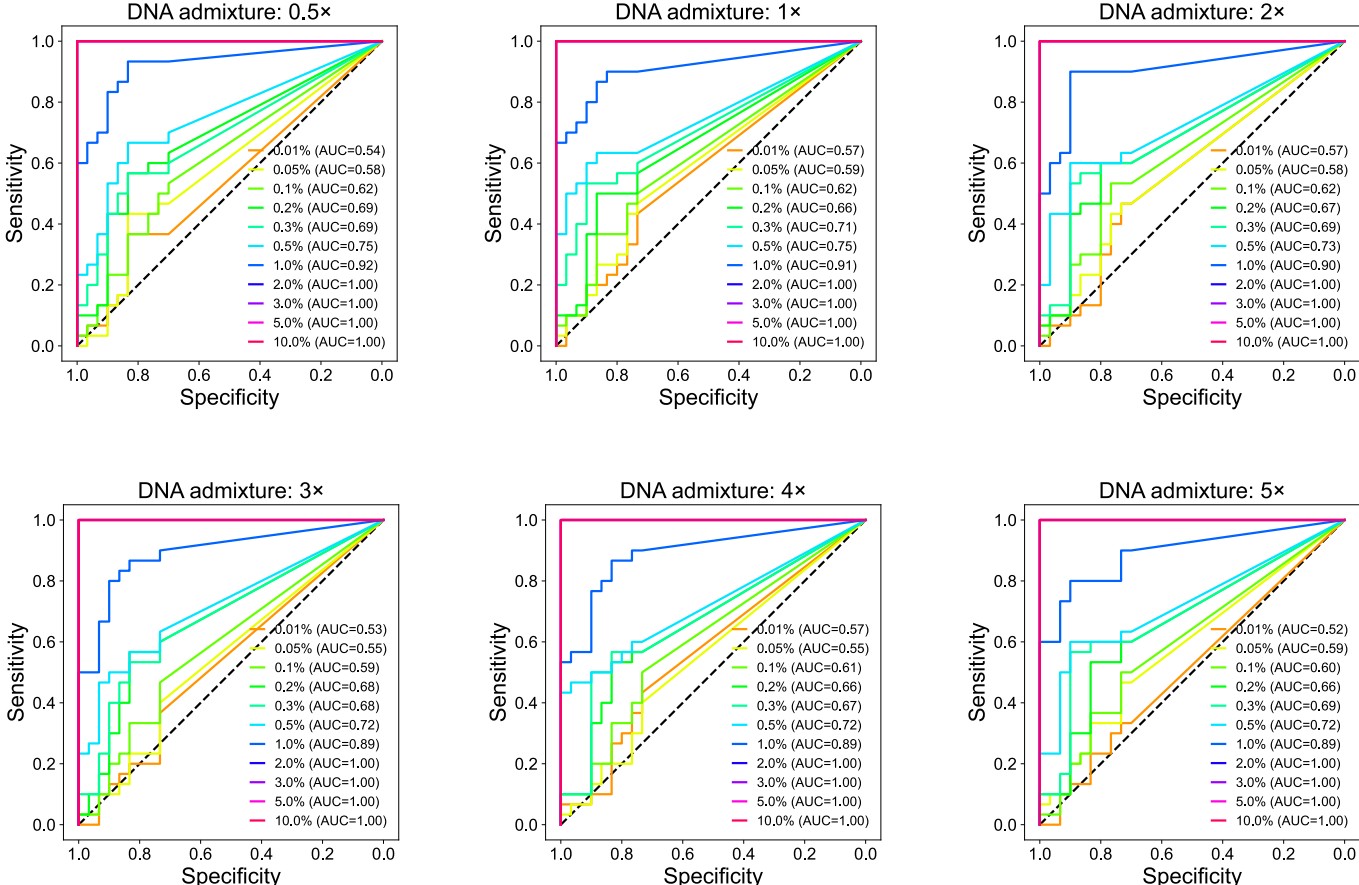

**Figure EV3. Performance of tumor DNA detection across sequencing depths and tumor fractions.**

ROC curves for differentiating diluted WGS data with and without tumor DNA across different sequencing depths and tumor fractions.

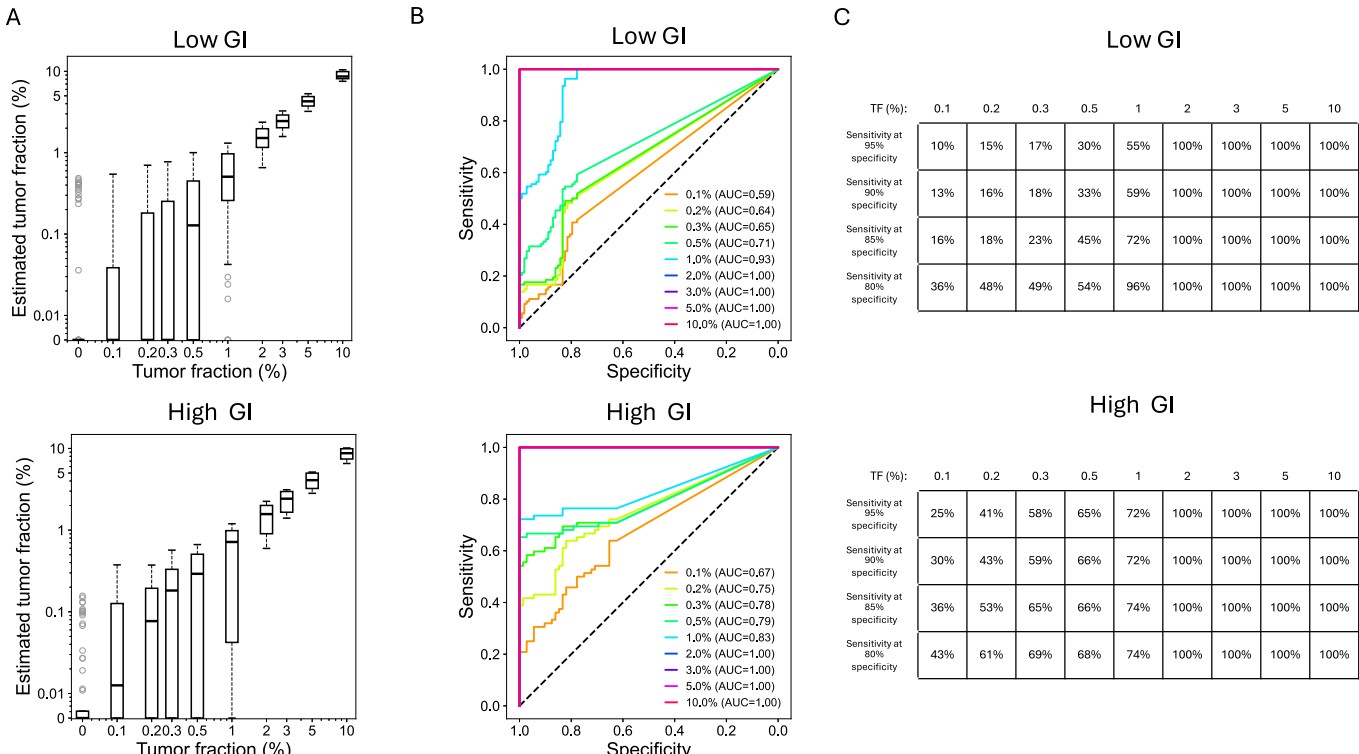

**Figure EV4. Analyses of in silico cancer cell line DNA dilution series with cancer types with high and low genomic instability (GI).**

The high GI group included four cell lines, including MKN-45 (gastric adenocarcinoma), KNS62 (non-small cell lung carcinoma), RMG-I (ovarian carcinoma), and SK-MEL-24 (melanoma). The remaining six cell lines were grouped into the low GI category, including AN3-CA (endometrial adenocarcinoma), H4 (glioma), LAMA-84 (chronic myeloid leukemia), SK-N-AS (neuroblastoma), SUIT-2 (pancreatic ductal adenocarcinoma), and 22Rv1 (prostate adenocarcinoma). (**A**) Boxplot of informCNA estimated TFs, with 72 and 108 estimations per tumor fraction in the high and low GI groups, respectively. The central line representing the median, the bounds of box indicating the interquartile range (IQR; 25% and 75% percentiles), and whiskers extend to the most extreme values within 1.5 times IQR. Outliers exceeding the whiskers are plotted individually to represent the full data range, including minimum and maximum values. (**B**) ROC curves for tumor DNA detection using informCNA estimated TF. (**C**) Sensitivity of informCNA tumor detection across different specificities.

## Overall samples

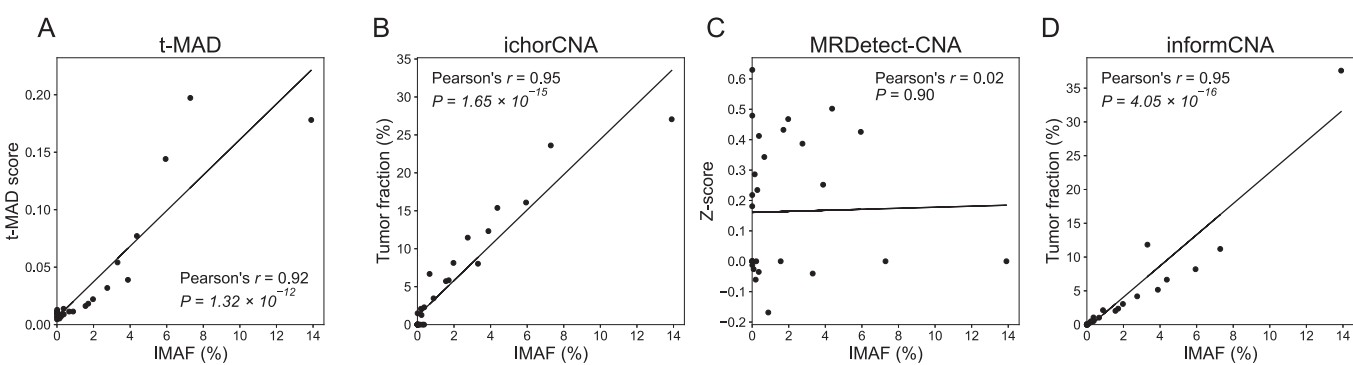

**Figure EV5.  Comparison of IMAF with copy-number–based ctDNA metrics.**

Correlation between IMAF values and the outputs of tMAD (**A**), ichorCNA (**B**), MRDetect-CNA (**C**), and informCNA (**D**) across all time points in patients with breast cancer.

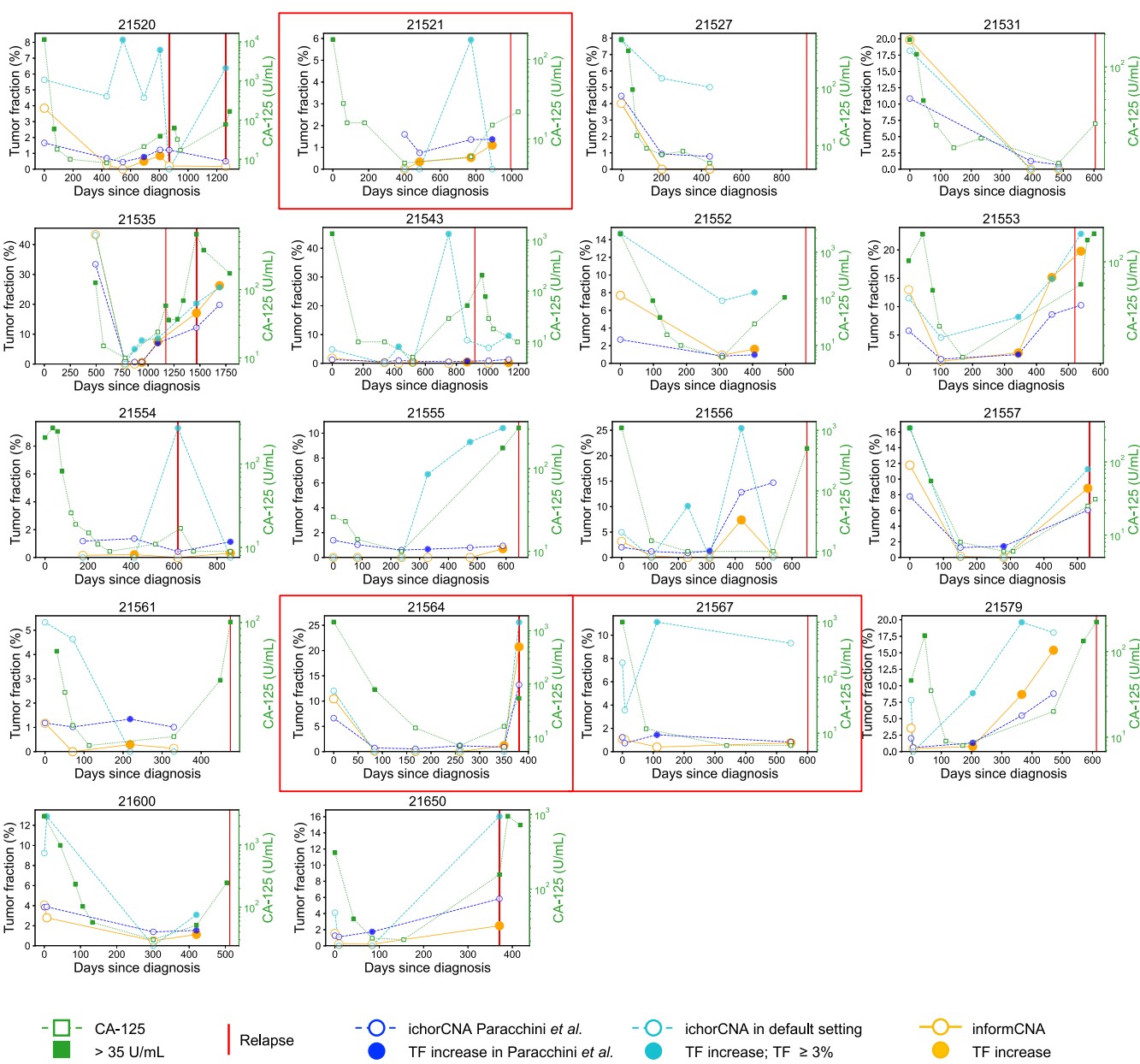

**Figure EV6. Performance of different assays in longitudinal monitoring of ovarian cancer patients undergoing chemotherapy.**

CA-125 levels (open green square) > 35 U/mL (filled green square) indicate detected recurrence. ctDNA tumor fractions (TFs) in plasma cfDNA are estimated either by ichorCNA in default setting (cyan circle) or with manual curation (blue circle; Paracchini et al), and by informCNA (yellow circle). In patient 21521, the manually curated ichorCNA TF at the second plasma time point showed a decreasing trend, whereas both CA-125 levels and informCNA estimated TF showed an increasing trend. In patient 21564, at the fourth plasma time point, the manually curated ichorCNA TF showed an increase, while both CA-125 and informCNA TF indicated a decrease. For patient 21567, both the manually curated and default ichorCNA TFs showed an increasing trend, in contrast to the decreasing trends observed in CA-125 and informCNA estimated TF. Notably, manually curated ichorCNA results also differed from those obtained using the default settings.

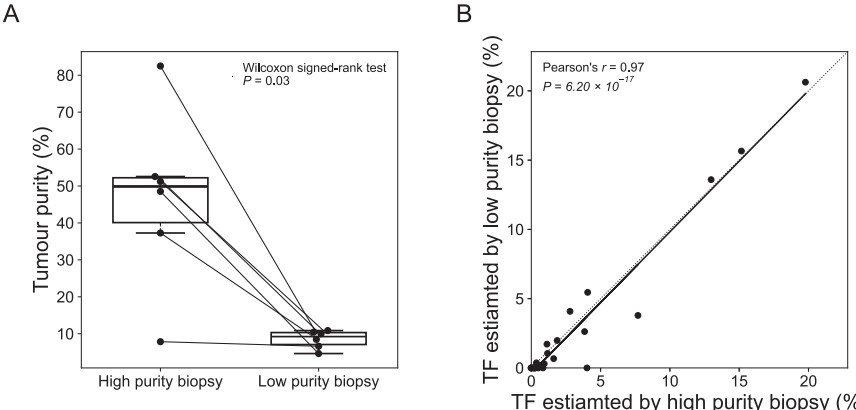

**Figure EV7.  ctDNA detection using references with low tumor purity.**

(A) Tumor purity of low-purity biopsies (median: 9.2%; range: 4.6 to 10.9%) and their high-purity compartments (median: 49.9%; range: 7.8 to 82.5%). The central line representing the median, the bounds of box indicating the interquartile range (IQR; 25% and 75% percentiles), and whiskers extend to the most extreme values within 1.5 times IQR. Outliers exceeding the whiskers are plotted individually to represent the full data range, including minimum and maximum values. (B) Correlation between TFs estimated by informCNA using either the high- or low-purity biopsy as the reference sample in same patient.

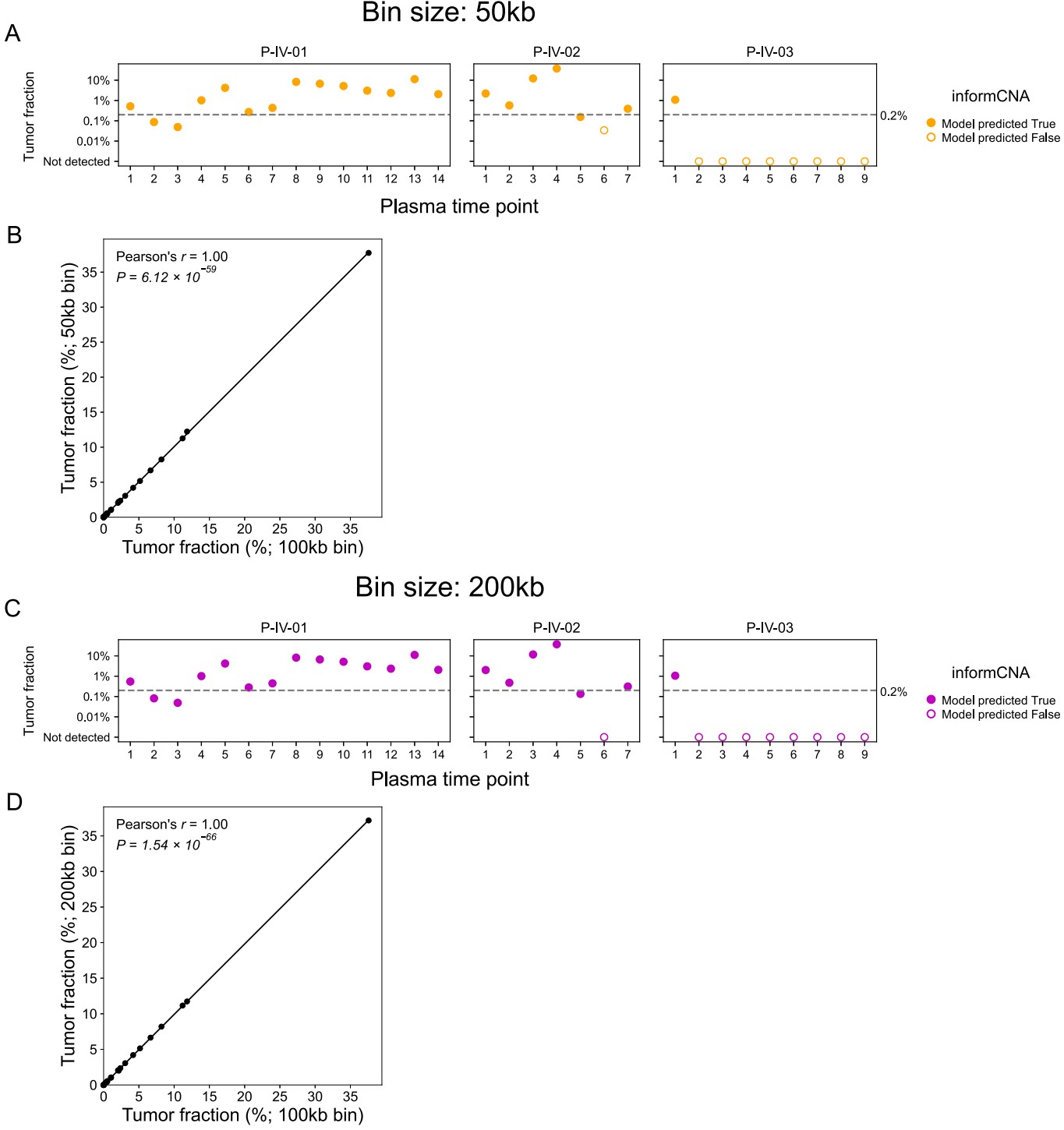

**Figure EV8.** ctDNA detection and tumor fraction correlation analyses using different window sizes.

(**A**) Tumor fraction estimated by informCNA using 50 kb bins. (**B**) Correlation between tumor fractions estimated by using 100 kb and 50 kb bins. (**C**) Tumor fraction estimated by informCNA using 200 kb bins. (**D**) Correlation between tumor fractions estimated by using 100 kb and 200 kb bins.

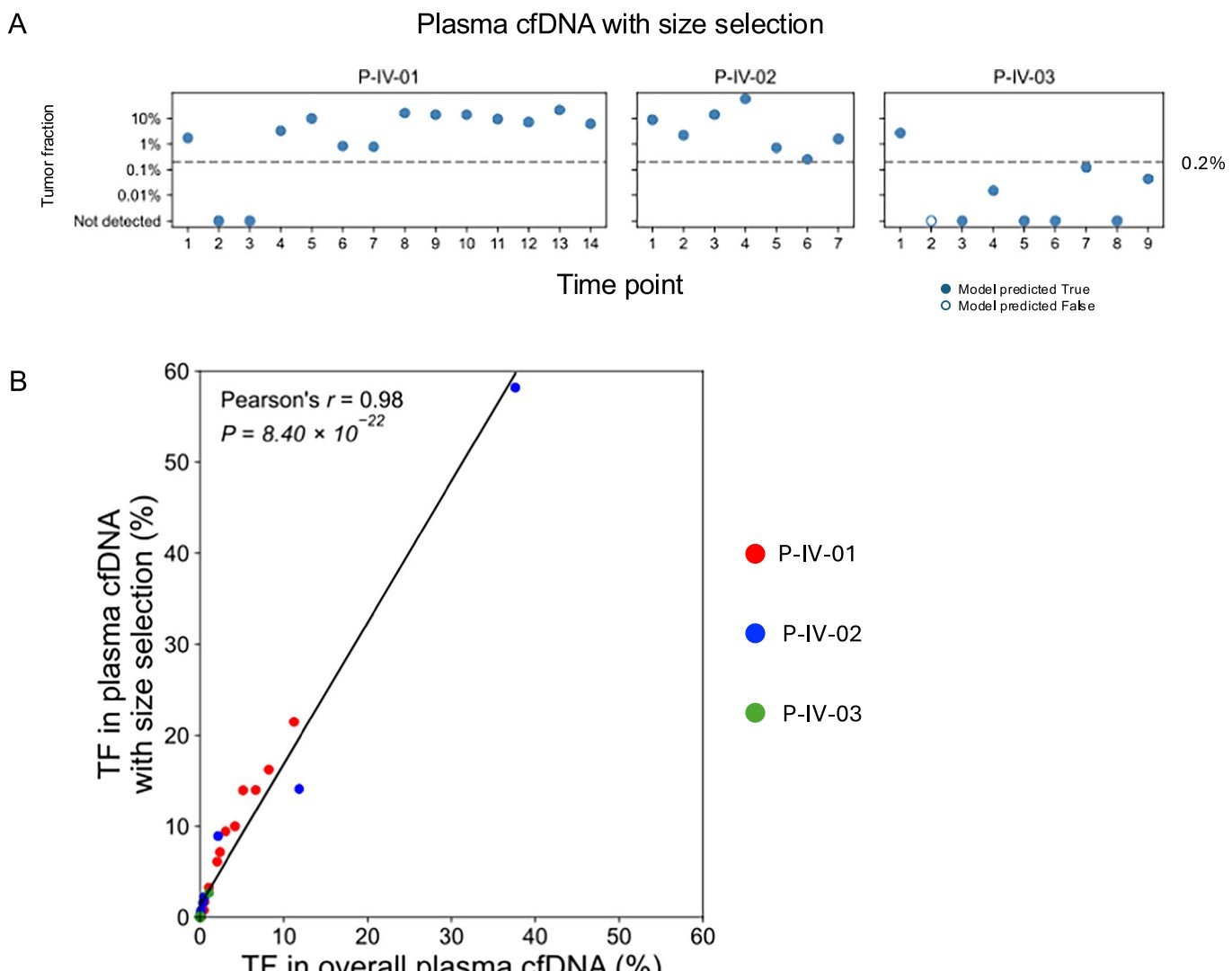

**Figure EV9. ctDNA detection and tumor fraction estimation using in silico size selection.**

(A) Tumor fraction estimated by informCNA using in silico size selection of short plasma cfDNA (< 150 bp). (B) Correlation between tumor fractions estimated from overall plasma cfDNA without size selection and from in silico size selected short cfDNA.

