## [Peer Review File · EMBO Molecular Medicine]

ctDNA monitoring using tumor-informed copy number analysis

Ze Zhou, Ros Cutts, Sarah Hrebien, Christina Zhang, Isaac García-Murillas, Woody Zhang, Alexander Frankell, Wendy Cooper, Amit Roshan, Nicholas Turner, Tommy Kaplan, Nitzan Rosenfeld, and Hui Zhao

Corresponding author: Hui Zhao (huizhao@qmul.ac.uk) , Nitzan Rosenfeld (nitzan.rosenfeld@cruk.cam.ac.uk)

Review Timeline:

Submission Date:	5th Aug 25
Editorial Decision:	10th Sep 25
Revision Received:	26th Nov 25
Editorial Decision:	19th Dec 25
Revision Received:	10th Feb 26
Accepted:	25th Feb 26

Editor: Lise Roth

Transaction Report:

10th Sep 2025

Dear Dr. Zhao,

Thank you for submitting your manuscript to EMBO Molecular Medicine, and please accept my apologies for the delay in getting back to you as one referee needed more time to provide a report. We have now received feedback from the three reviewers who agreed to evaluate your manuscript. As you will see from the reports below, they acknowledge the interest and high technical quality of the study but also make suggestions to improve the manuscript further. If you feel you can satisfactorily address the points listed by the referees, you may wish to submit a revised version of your manuscript.

Addressing the reviewers' concerns in full will be necessary for further considering the manuscript in our journal, and acceptance of the manuscript will entail a second round of review. EMBO Molecular Medicine encourages a single round of revision only and therefore, acceptance or rejection of the manuscript will depend on the completeness of your responses included in the next, final version of the manuscript. For this reason, and to save you from any frustrations in the end, I would strongly advise against returning an incomplete revision.

We are expecting your revised manuscript within three months, if you anticipate any delay, please contact us.

We require:

Additional information on source data and instruction on how to label the files are available

4) A .docx formatted letter INCLUDING the reviewers' reports and your detailed point-by-point responses to their comments. As part of the EMBO Press transparent editorial process, the point-by-point response is part of the Review Process File (RPF), which will be published alongside your paper.

5) A complete author checklist, which you can download from our author guidelines (<https://www.embopress.org/page/journal/17574684/authorguide#submissionofrevisions>). Please insert information in the checklist that is also reflected in the manuscript. The completed author checklist will also be part of the RPF.

6) All Materials and Methods need to be described in the main text using our 'Structured Methods' format. According to this format, the Methods section includes a Reagents and Tools Table (listing key reagents, experimental models, software and relevant equipment and including their sources and relevant identifiers) followed by a Methods and Protocols section describing the methods, ideally using a step-by-step protocol format. The aim is to facilitate adoption of the methodologies across labs. Please download and fill our Reagents and Tools Table template (.docx), which you can find in our author guidelines: <https://www.embopress.org/page/journal/14693178/authorguide#structuredmethods>.

7) Please note that all corresponding authors are required to supply an ORCID ID for their name upon submission of a revised manuscript.

8) It is mandatory to include a 'Data Availability' section after the Materials and Methods. Before submitting your revision, primary datasets produced in this study need to be deposited in an appropriate public database, and the accession numbers and database listed under 'Data Availability'. Please remember to provide a reviewer password if the datasets are not yet public (see <https://www.embopress.org/page/journal/17574684/authorguide#dataavailability>).

9) For data quantification: please specify the name of the statistical test used to generate error bars and P values, the number (n) of independent experiments (specify technical or biological replicates) underlying each data point and the test used to calculate p-values in each figure legend. The figure legends should contain a basic description of n, P and the test applied. Graphs must include a description of the bars and the error bars (s.d., s.e.m.). Please provide exact p values.

10) Our journal encourages inclusion of *data citations in the reference list* to directly cite datasets that were re-used and obtained from public databases. Data citations in the article text are distinct from normal bibliographical citations and should directly link to the database records from which the data can be accessed. In the main text, data citations are formatted as follows: "Data ref: Smith et al, 2001" or "Data ref: NCBI Sequence Read Archive PRJNA342805, 2017". In the Reference list, data citations must be labeled with "[DATASET]". A data reference must provide the database name, accession number/identifiers and a resolvable link to the landing page from which the data can be accessed at the end of the reference. Further instructions are available at .

11) We replaced Supplementary Information with Expanded View (EV) Figures and Tables that are collapsible/expandable online. EV Figures should be cited as 'Figure EV1, Figure EV2' etc... in the text and their respective legends should be included in the main text after the legends of regular figures.

12) The paper explained: EMBO Molecular Medicine articles are accompanied by a summary of the articles to emphasize the major findings in the paper and their medical implications for the non-specialist reader. Please provide a draft summary of your article highlighting

13) Author contributions: CRediT has replaced the traditional author contributions section because it offers a systematic machine readable author contributions format that allows for more effective research assessment. Please remove the Authors Contributions from the manuscript and use the free text boxes beneath each contributing author's name in our system to add specific details on the author's contribution. More information is available in our guide to authors.

Please also suggest a visual abstract to illustrate your article as a PNG file 550 px wide x 300-600 px high. A cropped portion of this image will serve as thumbnail for the table of content on our webpage.

16) As part of the EMBO Publications transparent editorial process initiative (see our Editorial at <http://embomolmed.embopress.org/content/2/9/329>), EMBO Molecular Medicine will publish online a Review Process File (RPF) to accompany accepted manuscripts.

In the event of acceptance, this file will be published in conjunction with your paper and will include the anonymous referee reports, your point-by-point response and all pertinent correspondence relating to the manuscript. Let us know whether you agree with the publication of the RPF and as here, if you want to remove or not any figures from it prior to publication. Please note that the Authors checklist will be published at the end of the RPF.

I look forward to receiving your revised manuscript.

Yours sincerely,

Lise Roth

***** Reviewer's comments *****

Referee #1 (Comments on Novelty/Model System for Author):

The work represents a strong technical advance. However, it is difficult to assess the medical impact, as the study was not powered to demonstrate that the tools provides better patient outcomes. However, determining medical impact is outside the scope of the current work and would be better suited in a follow-up manuscript.

Referee #1 (Remarks for Author):

In this manuscript, Zhou et al. present informCNA, a tumor-informed ctDNA quantification tool that uses a priori known copy number changes in a patient's tumor to improve ctDNA sensitivity. The tool was deployed in three different cancer types (breast, melanoma and ovarian), demonstrating broad applicability. The authors claim a limit of detection of 0.2%, which represents a ~10 fold improvement over ichorCNA, a tumor-naïve ctDNA detection tool and the current state of the art for low-pass, CNA-based ctDNA detection. The work is benchmarked to alternative tools and was deployed in patient vignettes to demonstrate potential clinical utility.

To my knowledge, there are very few (if any) easily deployable tumor-informed CNA-based ctDNA tools. Therefore, the work presented here is novel and of interest. If this software is maintained (which is very likely given the track record of this group), it has the potential to become a widely used tool.

Major comments on the manuscript:

1- Code

The code is easy to follow, well written and intuitive. Kudos to the authors.

2- informCNA rationale

Can the authors expand on the rationale behind the tools they used for informCNA? For example, the CNAs from the tumor are identified using custom python code, and not widely used software. It would be beneficial for the readers to understand the rationale behind such decisions.

3- informCNA benchmarking is incomplete

Additional benchmarking should be performed to answer the following questions:

3.1- What is the smallest alteration that can be detected (and at what TF) using incormCNA

3.2- Does performance of informCNA improve with coverage? What happens at 0.02x, 0.2x, 2x, 20x, etc.

3.3- Would buffy coat or a matched normal improve the performance of informCNA? Would deeper sequencing of the tumor?

3.4- the authors claim an LOD of 0.2%, but this was determined by finding the lowest simulated TF where 50% or more of samples had a TF>0. How does the informCNA LOD compare to ichorCNA, if measured the same way? Both LOD methods (median >0, and the methods described in ichorCNA) should be compared. This is hinted at in F2D and F3 but a clear head-to-head would be helpful.

4- informCNA model vs measured TF.

The authors state: "when informCNA is positivie, the Similarity Score component provides an estimate of the ctDNA fraction (or TF) in the sample." However, in F3, model predictions of "T" have 0% informCNA TFs and "-ve" informCNA ctDNA. The three rows of "informCNA TF", "Model prediction" and informCNA ctDNA" are not well described. Generally, the purpose of the SVM "+" or "-" classification was not clear to me.

Minor comments:

- 1- Line 59-61: is sWGS is typically less expensive than targeted or deep WGS. But is sWGS of cfDNA truly a cost effective option if it requires frequent blood draws (cost of staff and infrastructure) as opposed to a single, more costly assay?
- 2- Line 62-75: the authors should describe how ichorCNA and t-MAD work, as a general audience would not be able to follow why these tools are mentioned.
- 3- F3: Model prediction "T" should be spelled out
- 4- F3: the two different plots (ichorCNA on the top row, and the triangle blots on the bottom) are so small that readers cannot understand (A) the data and (B) the conclusion of the plot.
- 5- Line 241, 508: typo "MRDect-CNA"
- 6- The authors should consider a tumor-naive option for informCNA to provide a one-stop-shop for low pass CNA analysis, although this is merely a suggestion.

Referee #2 (Comments on Novelty/Model System for Author):

The paper is highly relevant and well-performed

Referee #2 (Remarks for Author):

This manuscript describes a very comprehensive analysis of tumor informed CNV analysis of ctDNA.

The idea of using shallow WGS data for ctDNA detection has been investigated in several studies but not reached sensitivity better than 3%. By using tumor information, the authors improve the sensitivity more than 10-fold, which is crucial for the clinical relevance. Shallow sequencing is considerably cheaper than deep sequencing and no normal tissue sequencing is needed. If no tumor sample is available a baseline sample with high tumor DNA content is proposed, which is obviously a good idea.

Many very relevant aspects, e.g. sensitivity and specificity across different cancers, bin sizes, performance compared to other methods etc. are evaluated ensuring robustness of informCNA.

There are several questions that would be of relevance to improve the paper:

-in many cases only FFPE will be available as tumor information source as baseline samples may also have low tumor content. The authors should mention this as a limitation or future focus area.

-The method presented in the paper has the potential to be implemented clinical at some point. Its main use might be for monitoring treatment response rather than early detection of recurrence since higher sensitivity would most likely be requested for this purpose.

- In the Discussion section line 326-338: informCNA achieves a LLOD of 0.2% ctDNA fraction (TF). It should be also noted here that this LLOD is only with 54% sensitivity at 80% specificity.

Also in the Abstract line 35: ... detect ctDNA down to 0.2% TF across multiple cancer types... - Not mentioning the sensitivity and specificity is a bit overstatement for the power of the tool.

- It is recommended to clearly state in the Discussion section that informCNA does not calculate a tumor fraction on its own but only a similarity score. One limitation of the tools is that the TF of the query sample depends on the accuracy of the tool used to calculate the TF of the reference sample.

The manuscript would benefit from a little more details on several aspects of the bioinformatics approach and pipeline:

-It is important that the code will be available and functional for users.

- To install dependencies for the Python scripts, the full versions need to be provided. Some are given but not all and not in the requirement.txt file.

- The workflow illustration for the codes, is not complete. The ctDNA detection using copy number significance analysis is missing, where the predict.py script should be run on the comparison result from the last step of the workflow. It is also recommended to add another step and show that TF of the reference needs to be estimated using other tools (e.g. ACE).

-an explanation of the weighted least square equation would be good

438: say clearly that reference sample means high tumor content sample.

445: more explanation is needed for the formula. What is x?

448: Explain reason for weighting sub chromosomal regions higher.

449: the weighting parameter is not intuitive. It seems to promote gains compared to loss. Is that correct? Please explain more.

The first part of model for significance evaluation seems logic. However, it is hard to follow what is going on with the p-value matrix and what the SVM model is doing. Please explain more.

The analysis of different bin size is performed on multiple samples from three breast cancer patients. The question is how similar their CNV landscapes are? The degree of genomic instability can be very different in different breast cancer patients and this can have big influence on the performance using different bin sizes.

Similar to this argument the reported sensitivity of 0.2% must also be very dependent on the level of genomic instability. Did the authors compare the sensitivity across different cancers with different GI?

Also the purity of the reference sample is an important parameter. In figure 5, Tf is shown to be robust even with low tumor content reference samples. It would be important with some more precise reporting on the needed tumor content. This is important since a frozen biopsies or baseline samples with detectable tumor content is often a limitation.

- The analysis done for the use of high-TF baseline (pre-treatment) plasma samples instead of tumor sample as a high TF reference is very limited. It can be a good addition to the study, if the authors run some additional analyses to validate their tool using pre-treatment plasma. Also it is not clear if the user wants to use pre-treatment plasma instead of tumor as reference, how

high should be the TF in the baseline plasma? To specify this limit, more analyses are needed.

MRDetect seem to run sub-optimal in the analysis since it does not detect samples with TF 1-10% (Figure 4D). In the introduction a sensitivity of MRDetect of 0.005% TF is quoted. One difference to MRDetect seems to be that they use CNVkit for this analysis that might explain this?

- From the methods section, it seems that ACE tool was used for the estimation of tumor purity. If the LLOD of this tool is 5% (according to the manuscript), doesn't this affect the analyses of reference tumors with TF below 5%?
- What does similarity score above 100 mean (Fig. S1 (A))? Is not 100 the max?
- Fig. S1(B): what is the reference here for the similarity score?
- Estimation of run time of the tool? Comparison with other tools like MRDetect/MRD-EDGE or some comment on it?
- System's requirement to run the pipeline (memory and CPU)?
- A list of abbreviations would be useful for the reader
- MRDect should be MRDetect

Referee #3 (Remarks for Author):

This manuscript describes informCNA, a tumor-informed copy number analysis framework for ctDNA detection from shallow WGS. The approach leverages patient-specific CNA profiles to improve sensitivity, claiming a limit of detection of 0.2% TF while maintaining specificity. The work is timely and relevant, addressing an important need for cost-effective, scalable ctDNA monitoring methods. Benchmarking against established CNA tools and clinical validation in ovarian, breast, and melanoma cohorts underscore its translational potential. Please find below some suggestions to further strengthen the paper and its claims.

1. Validation Design and Potential Confounders

- o The in silico dilution mixes cell-line WGS into cfDNA from healthy donors. This may introduce a confounder, as tumor tissue DNA and plasma cfDNA differ substantially in fragmentation patterns (an effect demonstrated previously by this group). It is not clear how this impacts informCNA's reported sensitivity.
- o Similarly, the in vitro dilution experiments are compelling, but details of the positivity cutoff ($\geq 0.2\%$ TF plus SVM positivity) could be explained more explicitly in the Results and figure legends.

2. Lower Limit of Detection

- o The authors highlight 0.2% TF detection as a key advance. However, at this level the AUC is only 0.67 with ~50% sensitivity at 80% specificity. Is this sufficient for clinical use? I suggest tempering the claim or contextualizing it more explicitly against clinical requirements for recurrence monitoring.

3. Copy Number Landscape Dependence

- o Performance likely depends on the breadth and amplitude of CNAs in the reference tumor. Cancers that are copy-quiet may be challenging to monitor, and the differential contribution of copy gains vs losses is not explored. The rationale for placing extra weight on focal lesions is mentioned but not justified with results. A stratified analysis of performance by CNA burden, or at least a discussion, would strengthen the paper.

4. Benchmarking Strategy

- o It is a strength that three methods (t-MAD, ichorCNA, MRDetect-CNA) were included. However, benchmarking appears to have been done only in clinical samples. Was there a reason these methods were not also applied to the in silico and in vitro series, which would have enabled a more controlled performance comparison?

5. Impact of Sequencing Depth

- o The method is positioned as cost-effective due to low-coverage sWGS. However, sequencing costs are steadily declining, and some may consider deeper sequencing. Have the authors examined whether informCNA performance improves with higher depth? A sensitivity analysis would be helpful.

6. Statistical Framework and Regularization

- o The Similarity Score relies on weighted least squares with LASSO regularization. The choice of penalty and its effect on detecting subtle CNA signals should be clarified. Since the approach aggregates many small deviations, how sensitive are results to regularization strength? A brief rationale and reference (e.g. DELFI, which also leverages subtle copy/fragmentomic signals) would help contextualize this choice.

7. Interpretation of Segment Correlation Figures

- o In the in vitro dilution series, the correlation structure (Fig. 3 lower panels) "lights up" as red with increasing TF. This seems to reflect segments beginning to correlate with each other, but the biological/statistical interpretation is not clear. Clarification in the text or legends would help readers.

Other Minor Concerns

- Figures: some multi-panel outputs (e.g. Fig. 3 significance plots) are difficult to follow without extended guidance.
- Discussion of potential confounding from clonal hematopoiesis CNAs in plasma-only mode is warranted.

The point-by-point responses to the reviewers' comments.

We thank the reviewers for their valuable and constructive comments. Our point-by-point responses to the comments are listed below.

Referee #1 (Comments on Novelty/Model System for Author):

The work represents a strong technical advance. However, it is difficult to assess the medical impact, as the study was not powered to demonstrate that the tools provides better patient outcomes. However, determining medical impact is outside the scope of the current work and would be better suited in a follow-up manuscript.

Response: We thank the reviewer for thorough evaluation of our manuscript and for providing us with overall positive and encouraging comments.

Referee #1 (Remarks for Author):

In this manuscript, Zhou et al. present informCNA, a tumor-informed ctDNA quantification tool that uses a priori known copy number changes in a patient's tumor to improve ctDNA sensitivity. The tool was deployed in three different cancer types (breast, melanoma and ovarian), demonstrating broad applicability. The authors claim a limit of detection of 0.2%, which represents a ~10 fold improvement over ichorCNA, a tumor-naïve ctDNA detection tool and the current state of the art for low-pass, CNA-based ctDNA detection. The work is benchmarked to alternative tools and was deployed in patient vignettes to demonstrate potential clinical utility.

To my knowledge, there are very few (if any) easily deployable tumor-informed CNA-based ctDNA tools. Therefore, the work presented here is novel and of interest. If this software is maintained (which is very likely given the track record of this group), it has the potential to become a widely used tool.

Major comments on the manuscript:

1- Code

The code is easy to follow, well written and intuitive. Kudos to the authors.

Response: We thank the reviewer for detailed review of our codes.

2- informCNA rationale

Can the authors expand on the rationale behind the tools they used for informCNA? For example, the CNAs from the tumor are identified using custom python code, and not widely used software. It would be beneficial for the readers to understand the rationale behind such decisions.

Response: We thank the reviewer for raising this question. The fundamental steps of CNA detection tools are largely similar: dividing genome into bins, performing bias correction (for GC content, mappability, etc.) and segmenting the data to infer copy number profiles. Among which, the core computation step is segmentation. Widely used software for copy number profiling, including QDNAseq (Scheinin *et al.* 2014), CNVkit (Talevich *et al.* 2016), and ichorCNA (Adalsteinsson *et al.* 2017), are all based on the same principles but are designed for different applications, namely for low-pass WGS, targeted sequencing and cfDNA low-pass WGS, respectively. These tools were all developed around a decade ago. For the segmentation methods, QDNAseq uses circular binary segmentation (CBS), ichorCNA uses

Hidden Markov Model (HMM), and CNVkit uses both plus Fused Lasso segmentation. These methods can be sensitive to noise, leading to false positives, and can be time-consuming. More recent tools such as ACE (Poell *et al.* 2019) and Rascal (Sauer, *et al.* 2021) build upon the same framework. ACE is often used downstream of QDNAseq to infer absolute CN states and purity, while Rascal is typically applied after ACE for final refinement. Their fundamental approaches to bias correction and segmentation still originate from QDNAseq. Moreover, none of these tools are able to detect both large copy-number alterations and focal copy-number alterations. Regarding the use of custom Python code rather than developed tools, e.g., *scipy* for data processing, *ruptures* for segmentation, and *cvxpy* for Similarity Score fitting, we believe that transparency is important for methods with potential clinical applications. By developing *informCNA* from the ground up, we are able to fully understand and control every parameter at each step, including read-depth estimation, segmentation, and downstream analysis. The expertise gained in building *informCNA* will further support the development of a tumor-naïve CNA assay in future studies.

We also listed all external tools used in *informCNA*.

Software	Rationale
BWA	A widely used, accurate, and efficient aligner for mapping sequencing reads to reference genomes.
Picard	A comprehensive toolkit for processing and manipulating high-throughput sequencing alignment data.
Samtools	The standard suite for viewing, sorting, indexing, and manipulating SAM/BAM files.
GenMap	Enables ultra-fast computation of genome mappability.
Python	A versatile and widely adopted programming language in bioinformatics.
pysam	A Python interface for reading, writing, and manipulating SAM/BAM files efficiently.
pyBigWig	A Python interface for fast and efficient access to BigWig and BigBed files, and supports writing and manipulation.
cvxpy	A modeling language for solving convex optimization problems in Python.
matplotlib	A comprehensive library for creating visualizations in Python.
numpy	The core package for large, multi-dimensional arrays and matrices computation in Python.
pandas	A powerful and flexible package for data frame analysis and manipulation in Python.
ruptures	A Python package providing efficient algorithms for change-point detection based on linearly penalized segmentation methods.
scipy	A package providing fundamental algorithms for mathematics, science, and engineering in Python.
scikit_posthocs	A package offering a variety tests for pairwise multiple comparisons in statistical analyses in Python.

scikit-learn	A machine learning library built on SciPy, for data mining, analysis, and model develop in Python.
--------------	--

We have now strength it in the revised manuscript (pages 15 to 16, lines 436 to 437).

3- informCNA benchmarking is incomplete

Additional benchmarking should be performed to answer the following questions:

3.1- What is the smallest alteration that can be detected (and at what TF) using informCNA.

Response: We set 2 Mb of alignable genomic region as the threshold for the minimum detectable CNA in the code. To demonstrate the lowest TF at which the minimum CNA (2 Mb) could be detected, we conducted an *in silico* dilution series to evaluate the ability of informCNA to identify both gain and loss regions. We selected chromosome 1, containing 20,000 bins, from a buffy coat sample without any CNAs. Next, we artificially introduced one 2 Mb gain and one 2 Mb loss region at a random genomics location, separated by a 1 Mb copy-neutral region them. Copy number log₂ ratio changes induced by TF were simulated by adding corresponding values to all bins within each region (e.g. 2% TF was represented as +0.02 in the gain region and -0.02 in the loss region). We simulated a TF range from 1% to 20% and with ten replicates at each TF, totalling 90 simulated samples. Segmentation analysis was then applied to the resulting dataset. As shown in Figure R1, simulations with TF > 10% yielded consistent detection across all replicates, indicating that 10% TF (or 0.1 log₂ ratio) is required for reliable detection of 2 Mb segments.

Figure R1. Copy number aberration segment detection across different tumor fractions.

Given that the bin size is 100 kb, we further challenged informCNA to detect focal CNAs ranging from 100 kb to 500 kb, using a fixed log₂ copy ratio difference of 0.5 relative to the background copy-neutral bins. The simulation followed the same procedure described above, with ten replicates for each CNA length. As shown in Figure R2, informCNA detected all ten replicates in CNA segments ≥ 300 kb, indicating that 300 kb represents the smallest reliably detectable CNA size.

Figure R2. Copy number aberration segment detection for different lengths of CNA segments.

Of note, the identification of CNA alterations could also be influenced by multiple factors, including the TF raised by the reviewer, the genomic size of CNAs, copy number status, and genomic instability (Reviewer #2, comment 8). For example, chromosomal arm-level CNAs spanning hundreds of Mb can be reliably identified, while focal CNAs of smaller size but higher copy number amplitude are also easier to detect. In addition, adjacent regions showing both copy number gains and losses, with greater relative amplitude differences than their deviations from copy-neutral states, can also facilitate CNA detection. We have now included this in the discussion section of revised manuscript (page 14, lines 376 to 379).

3.2- Does performance of informCNA improve with coverage? What happens at 0.02x, 0.2x, 2x, 20x, etc.

Response: We thank the reviewer for the comment. In our *in silico* down-sampling analysis in the previous manuscript, we applied Copy Number Similarity Analysis to tumor WGS ranging from 0.1x to 5x coverage. As shown in Figure EV2, informCNA can extract highly concordant CNA information at $\geq 0.2x$, with average Similarity Score $>99\%$. As noted by the reviewer, at 0.02x coverage, the extremely low number of reads within bins (~25 reads per 100 kb bin) would result in increased variability in coverage estimation and poor Similarity Score.

To further evaluate the effect of sequencing depth on copy number profiling, we performed down-sampling on the cancer cell line 22Rv1. As shown in Figure R3A, the copy number \log_2 ratios at 0.02x exhibited broad variability across the genome, leading to an increased risk of CNA segments overestimation. Figure R3B illustrates the kernel density distributions of genome wide copy ratio from 0.02x to 20x coverage. Three distinct peaks at approximately 1, 0, and 0.5 (x-axis) correspond to genetic regions with copy number loss, neutral, and gain, respectively. At 0.02x coverage, these peaks are blurred but become clearly distinguishable as coverage increases to 0.2x (Figure R3B).

This trend is further reflected in the standard deviation (SD) of the copy ratios across bins: 0.02x has the highest standard deviation (SD: 0.555), followed by 0.2x (SD: 0.325), 2x (SD: 0.297), and 20x (SD: 0.294). The reduction in SD is more pronounced at low coverage, while differences between 2x and 20x are minimal.

This observation aligns with the law of large numbers, whereby increasing sequencing depth reduces read count variability and leads to more stable copy number estimates. However, as shown in Figure R3B, further increasing tumor sequencing depth (from 2x to 20x) yields minimal improvement in copy number profiling precision (SD from 0.297 to 0.294) but substantially increases sequencing cost.

Figure R3. (A) Copy number profile distribution of 22Rv1 at 0.02x, 0.2x, 2x and 20x coverage using informCNA. (B) Kernel density of copy number in every bin across different sequencing depth.

3.3- Would buffy coat or a matched normal improve the performance of informCNA? Would deeper sequencing of the tumor?

Response: Unlike MRDetect-CNA, which leverages high-depth (>30x) sequencing of tumor and matched normal samples (e.g., buffy coat) to generate high-resolution (single base pair) CNA calls to guide ctDNA detection (Zviran *et al*, 2020), informCNA was designed to utilize sWGS of both high-TF reference and low-TF query plasma cfDNA samples for frequent disease monitoring, thereby facilitating timely treatment decisions.

As shown in Figure R4, informCNA detected no CNA segment in buffy coat samples from breast cancer patients ($n = 3$). Given that normal tissues are typically copy-neutral, additional sequencing of a normal sample generally does not provide further information about tumor CNAs in the sWGS setting. In addition, omitting normal sample sequencing helps maintain the cost-effectiveness of the assay.

Figure R4. Copy number profile of buffy coat samples from three breast cancer patients using informCNA.

Regarding deeper sequencing of tumor samples, as we noted in our answer for Reviewer #1, comment 3.3, higher sequencing depth yields greater read counts, which can provide a more robust copy number profile with reduced variation (consistent with the law of large numbers). However, as demonstrated in our previous manuscript, informCNA already achieves high concordance in CNA detection across tumor samples at various sequencing depths (>0.2x with > 99% Similarity Scores). Further increasing tumor sequencing depth does not substantially improve performance. We have now strengthened this point in the revised manuscript (page 7, lines 162 to 163).

3.4- the authors claim an LOD of 0.2%, but this was determined by finding the lowest simulated TF where 50% or more of samples had a TF>0. How does the informCNA LOD compare to ichorCNA, if measured the same way? Both LOD methods (median >0, and the methods described in ichorCNA) should be compared. This is hinted at in F2D and F3 but a clear head-to-head would be helpful.

Response: We thank the reviewer for the insightful comment. To assess the LOD of ichorCNA, we applied the tool to our *in silico* dilution series, generated by mixing tumor DNA from cancer cell lines with healthy plasma cfDNA. Given that the reported LOD of ichorCNA is 3%, we synthesized DNA mixture samples with TFs of 0%, 1%, 3%, 4%, 5%, 7%, and 10% at 1x coverage. Each TF level included 10 cancer cell lines with 3 replicates, totalling 210 samples.

As shown in Figure 5R, ichorCNA-estimated TFs exhibited an overestimated trend at low TF levels. Samples with 0% spiked-in tumor DNA showed a median TF of 4.4% (IQR: 0.0-5.1%); 1% showed 4.4% (IQR: 0.0-5.1%); 3% showed 4.6% (IQR: 3.6-5.3%); 4% showed 4.9% (IQR: 4.2-5.6%); and 5% showed 5.0% (IQR: 4.6-5.8%). All groups with spiked-in tumor DNA $\leq 4\%$ had median estimates above 4% TF.

In the ROC curves analysis, the 4% spiked-in tumor DNA samples yielded an AUC of 0.66 (informCNA achieved an AUC 0.67 at 0.2% TF), with approximately 30% sensitivity (informCNA:54%) at 80% specificity. Together, these results suggest that ichorCNA demonstrated an effective LOD of approximately 4% TF in our simulation dataset, when measured in the same way as informCNA.

Figure R5. Tumor fraction (TF) estimation for *in silico* dilution series using ichorCNA. (A) Boxplots of estimated TFs across different dilution groups. (B) Area under the ROC curves across varying TF levels. (C) Table showing sensitivity at different specificity (ranging from 80% to 95%).

Figure 2 informCNA results were attached for comparison:

4- informCNA model vs measured TF.

The authors state: "when informCNA is positive, the Similarity Score component provides an estimate of the ctDNA fraction (or TF) in the sample." However, in F3, model predictions of "T" have 0% informCNA TFs and "-ve" informCNA ctDNA. The three rows of "informCNA TF", "Model prediction" and informCNA ctDNA" are not well described. Generally, the purpose of the SVM "+" or "-" classification was not clear to me.

Response: We thank the reviewer for the comments. informCNA consists of two modules: (1) Copy Number Similarity Analysis, which estimates TF of the query samples, and (2) Copy Number Significance Analysis, which employs an SVM model to classify samples as either containing ctDNA (True) or not (False). The final result for each query sample, ctDNA positive (+ve) or negative (-ve), integrates these two outputs to ensure high specificity in ctDNA detection. Specifically, a sample is considered ctDNA-positive only if the estimated TF exceeds the limit of detection (0.2%) and the SVM model predicts True. We have now emphasized this point in Figure 3 caption and in the revised manuscript (page 8, lines 223 to 225).

Minor comments:

1- Line 59-61: is sWGS is typically less expensive than targeted or deep WGS. But is sWGS of cfDNA truly a cost effective option if it requires frequent blood draws (cost of staff and infrastructure) as opposed to a single, more costly assay?

Response: We thank the reviewer for the insightful comment. We agree that a single, although costly, assay for postoperative ctDNA detection can precisely identify minimal residual disease (MRD) and enable accurate risk stratification (Diaz, 2025; Widman *et al.*, 2024; Zviran *et al.*, 2020). Such assays are particularly valuable for identifying ctDNA-negative patients who are likely to experience long-term recurrence-free survival without requiring additional adjuvant therapy (Tie *et al.*, 2022).

However, for ctDNA-positive patients who proceed with adjuvant therapy, a cost-efficient and rapid-turnaround assay is essential to enable frequent monitoring of treatment response, early detection of recurrence, and real-time guidance of therapeutic decisions. informCNA, which relies solely on shallow whole-genome sequencing (sWGS) data, is designed to address this clinical need. Its low sequencing cost, simple workflow, and short processing time make it particularly suitable for longitudinal monitoring, where multiple tests are required over the course of treatment.

Additionally, for ctDNA-negative patients who continue with active surveillance, false negatives, or tumour evolution may still result in progression at a later date....

While repeated testing inevitably adds operational costs (e.g., blood collection and staff time), the sequencing cost per informCNA test is markedly lower than that of targeted panels or deep WGS, owing to its shallow coverage and streamlined workflow. As a result, even with frequent sampling, sWGS remains a cost-effective strategy for real-time monitoring. Importantly, the two approaches serve distinct but complementary clinical roles: single high-cost assays are optimal for initial MRD detection and risk stratification, whereas sWGS-based assays provide a practical solution for repeated, rapid assessment during active surveillance or adjuvant therapy.

We have now incorporated this discussion in the revised manuscript (page 4, lines 109–114; page 14, lines 338–394).

2- Line 62-75: the authors should describe how ichorCNA and t-MAD work, as a general audience would not be able to follow why these tools are mentioned.

Response: We thank the reviewer for pointing this out. ichorCNA, one of the most widely used tumor-naïve tools, employs a Hidden Markov Model to predict CNA segments and a Bayesian mixture model to estimate the TF. t-MAD, a tool previously developed by our team, detects tumor DNA using the denoised median absolute deviation (MAD) of genome-wide copy number distribution relative to the copy-neutral state. We have now included this in the introduction section of revised manuscript (pages 3 to 4, lines 85 to 91).

3- F3: Model prediction "T" should be spelled out.

Response: True (T) indicates tumor-specific signal detected, and False (F) indicates no tumor-specific signal detected by the SVM model. We have revised the Figure 3 caption and clarified in the revised manuscript (page 30, lines 829 to 833).

4- F3: the two different plots (ichorCNA on the top row, and the triangle blots on the bottom) are so small that readers cannot understand (A) the data and (B) the conclusion of the plot.

Response: We thank the reviewer for the suggestions. As shown in Figure R6, we have added a horizontal line between the ichorCNA and informCNA results to improve visual separation. Additionally, we increased the spacing between Figures 3A and 3B for clearer distinction. Vector graphics were used to ensure all details (e.g., segment coordinates and p-values) remain clear upon zooming. Because the journal's submission system may reduce figure resolution during Word-to-PDF conversion, we submitted the response letter and figures in

PDF format to avoid this issue. The figure caption has also been clarified in the revised manuscript (page 30, lines 829 to 846).

Figure R6. Revised version of Figure 3 with clearer separation between different parts.

5- Line 241, 508: typo "MRDetect-CNA"

Response: We have corrected the typo.

6- The authors should consider a tumor-naïve option for informCNA to provide a one-stop-shop for low pass CNA analysis, although this is merely a suggestion.

Response: We thank the reviewer for the suggestion. We plan to build upon our experience and expertise gained through the development of informCNA to create another tumor-naïve assay in a future study. This expertise includes methods for coverage calculation, normalization, and segmentation of sequencing data, which will serve as a strong foundation for developing such an approach.

Referee #2 (Comments on Novelty/Model System for Author):

The paper is highly relevant and well-performed.

Response: We thank the reviewer for the overall positive and constructive comments.

Referee #2 (Remarks for Author):

This manuscript describes a very comprehensive analysis of tumor informed CNV analysis of ctDNA.

The idea of using shallow WGS data for ctDNA detection has been investigated in several studies but not reached sensitivity better than 3%. By using tumor information, the authors improve the sensitivity more than 10-fold, which is crucial for the clinical relevance. Shallow sequencing is considerably cheaper than deep sequencing and no normal tissue sequencing is needed. If no tumor sample is available a baseline sample with high tumor DNA content is proposed, which is obviously a good idea.

Many very relevant aspects, e.g. sensitivity and specificity across different cancers, bin sizes, performance compared to other methods etc. are evaluated ensuring robustness of informCNA.

There are several questions that would be of relevance to improve the paper:

1. In many cases only FFPE will be available as tumor information source as baseline samples may also have low tumor content. The authors should mention this as a limitation or future focus area.

Response: We thank the reviewer for raising this point. Routine tumor biopsies are commonly formalin-fixed and paraffin-embedded (FFPE). Although FFPE samples show high concordance with fresh-frozen tumor tissues, they may still be limited by low tumor cellularity and copy-number artifacts (Basyuni et al, 2024). We have included the discussion in the revised manuscript (page 15, lines 424 to 426).

2. The method presented in the paper has the potential to be implemented clinical at some point. Its main use might be for monitoring treatment response rather than early detection of recurrence since higher sensitivity would most likely be requested for this purpose.

Response: We thank the reviewer for this insightful comment. We agree that the primary clinical utility of informCNA lies in monitoring treatment response, where changes in ctDNA fraction reflect tumor burden dynamics over time. As shown in Figure 1A, the principles underlying treatment response monitoring and early recurrence detection are closely related, as both rely on ctDNA as a biomarker of disease burden.

While high-sensitivity minimal residual disease (MRD) assays based on deep sequencing are well suited for early recurrence detection, their high cost, often several thousand pounds per test, limits their feasibility for frequent use. In contrast, informCNA is a cost-effective, fast-turnaround and scalable alternative that relies solely on shallow whole-genome sequencing (sWGS), enabling regular, real-time assessment of treatment efficacy.

We therefore envision informCNA as a complementary tool within current liquid biopsy diagnostic and prognostic framework, particularly valuable in the recurrent disease or

adjuvant therapy setting, where ongoing, rapid, cost-efficient monitoring is needed. We have now incorporated this point in the revised manuscript (page 14, lines 386 to 398).

3. In the Discussion section line 326-338: informCNA achieves a LLOD of 0.2% ctDNA fraction (TF). It should be also noted here that this LLOD is only with 54% sensitivity at 80% specificity.

Also in the Abstract line 35: ... detect ctDNA down to 0.2% TF across multiple cancer types.
- Not mentioning the sensitivity and specificity is a bit overstatement for the power of the tool.

Response: We thank the reviewer for the suggestions. We have included the 54% sensitivity at 80% specificity in the discussion section of revised manuscript (page 13, lines 363 to 364).

4. It is recommended to clearly state in the Discussion section that informCNA does not calculate a tumor fraction on its own but only a similarity score. One limitation of the tools is that the TF of the query sample depends on the accuracy of the tool used to calculate the TF of the reference sample.

Response: We agree with the comment. Strategically, informCNA does not directly calculate the TF of a query sample. Instead, it provides a Similarity Score relative to the TF of reference samples. Consequently, accurate TF estimation requires sufficient tumor DNA in the reference biopsies, with CNA segments detectable in a tumor-naïve assay. Typically, a TF of >3-5% is required for reliable performance. We have included the discussion in the revised manuscript (page 15, lines 432 to 435).

5. The manuscript would benefit from a little more details on several aspects of the bioinformatics approach and pipeline:

5.1 It is important that the code will be available and functional for users.

Response: We agree with the comment, and the source code of informCNA will be publicly available on GitHub under our group repository (<https://github.com/nrlab-CRUK/informCNA>) after the patent filing.

5.2 To install dependencies for the Python scripts, the full versions need to be provided. Some are given but not all and not in the requirement.txt file.

Response: We thank the reviewer for this helpful suggestion. We have revised the *requirements.txt* file to include package version information consistent with that reported in the Methods section.

5.3 The workflow illustration for the codes, is not complete. The ctDNA detection using copy number significance analysis is missing, where the *predict.py* script should be run on the comparison result from the last step of the workflow. It is also recommended to add another step and show that TF of the reference needs to be estimated using other tools (e.g. ACE).

Response: We thank the reviewer for the suggestions. We have added additional steps and annotations to the revised workflow illustration, showing that the TF of reference samples should be estimated using tumor-naïve tools (e.g., ACE or ichorCNA), and that the SVM prediction results for copy number significance analysis should be generated using *predict.py*

(Figure R7).

Tumor-informed cfDNA copy number analysis (informCNA)

1. The TF of reference sample needs to be estimated using tumor-naïve tools (e.g., ACE or ichorCNA).
2. The SVM prediction result for Copy Number Significance Analysis needs to be obtained via *predict.py*.

Figure R7. Illustration of informCNA workflow.

5.4 an explanation of the weighted least square equation would be good

Response: We previously presented the weighted least squares (WLS) equation, which illustrates how segment-specific weights are integrated into the Similarity Score calculation. These weights enable informCNA to place greater emphasis on more informative CNA segments with higher amplitude, thereby enhancing the detection of tumor-specific CNA signals. In the revised manuscript, we have refined this explanation for improved clarity (page 19, 504 to 507).

5.5 438: say clearly that reference sample means high tumor content sample.

Response: We have clarified that the reference sample refers to a high-tumor-content sample and have revised the manuscript accordingly (page 5, line 145).

5.6 445: more explanation is needed for the formula. What is x ?

Response: x is the value of copy number Similarity Score, we have clarified it in the revised manuscript (page 28, lines 785).

5.7 448: Explain reason for weighting sub chromosomal regions higher.

Response: We believe the sub-chromosomal regions referred by the reviewer are segments that exhibit altered copy number states compared with their adjacent regions. These regions are assigned higher weights because they are more informative than background regions with similar copy number states. For example, we include focal copy number aberrations, even though they are narrow in breadth, because their large amplitude differences offer stronger evidence for detecting potential CNAs in the query sample. We have clarified this rationale in the revised manuscript (page 20, lines 514 to 516).

5.8 449: the weighting parameter is not intuitive. It seems to promote gains compared to loss. Is that correct? Please explain more.

Response: We agree with the reviewer's interpretation. informCNA leverages copy number alterations across different genomic regions. Copy number gains in tumor tissue may range from 3 to 4 copies or even higher (≥ 5 copies), while copy number losses may range from 1 to 0 copy. As noted above, regions with larger amplitude differences provide more informative signals and are therefore prioritized in the analysis.

6. The first part of model for significance evaluation seems logic. However, it is hard to follow what is going on with the p-value matrix and what the SVM model is doing. Please explain more.

Response: In the previous manuscript, we have developed a segment comparison analysis to detect ctDNA by applying statistical pairwise comparisons across all segments within the query sample. The resulting p-values quantify the segment-level copy number differences. We organized these p-values in a lower-triangular matrix (shown in Figure 3), each dot represents a Dunn's test result. The color scale transitions from blue to red, indicating increasing level of statistical significance. The prominent red region in the lower-left corner corresponds to comparisons between the lowest- to highest- ranked segments, which exhibit the largest copy number differences. Additional information has been included in the revised manuscript (page 30, lines 834 to 838).

7. The analysis of different bin size is performed on multiple samples from three breast cancer patients. The question is how similar their CNV landscapes are? The degree of genomic instability can be very different in different breast cancer patients and this can have big influence on the performance using different bin sizes.

Response: We agree with the reviewer's comment that bin size can affect the detection of CNAs. Small CNAs may be missed when larger bins are used, whereas smaller bins can capture more random noise. Therefore, bin size is generally considered as a technical trade-off between detection resolution and sequencing coverage.

We also noted that Peter Sloombeek *et al.* recently presented a poster introducing an sWGS-adjusted Genomic Instability Score (GIS) derived from ichorCNA-based copy number analysis. However, as no manuscript or methodological details have yet been published, we were unable to compute the GIS in our study.

To assess the degree of genomic instability in our cohort, we applied ichorCNA to sWGS data from tumor tissue samples of three patients and evaluated TF and ploidy. As shown in Figure R8, the tumor tissues from these three breast cancer patients exhibited high genomic instability, with ploidy values ranging from 2.4 to 3.0.

Figure R8. Results of ichorCNA analysis on tumor biopsy samples obtained from three patients with stage IV breast cancer.

As also discussed in response to Reviewer #2 comment 8, genomic instability can vary across different cancer types and may further enhance detection performance in cancers with high genomic instability. Therefore, further optimization of the bin size may be required for optimal performance when informCNA is trained using data from specific cancer types. We have included this discussion in the revised manuscript (page 13, lines 380 to 383).

8. Similar to this argument the reported sensitivity of 0.2% must also be very dependent on the level of genomic instability. Did the authors compare the sensitivity across different cancers with different GI?

Response: We thank the reviewer for the insightful comment. We agree that genomic instability (GI) is a “facilitating characteristic” in the generation of cancer biomarkers (Andor et al. 2017). A higher level of GI can lead to a greater number of CNA segments with higher CNA amplitude, thereby facilitating ctDNA detection in both tumor-naïve and tumor-informed settings. Cancer types with high GI include breast, lung, oesophageal, stomach, colorectal, ovarian, bladder, and melanoma (Steele et al. 2022). Accordingly, we divided the cancer cell lines ($n = 10$) in the *in silico* dilution experiments into two categories: high GI ($n = 4$) and low GI ($n = 6$). The high GI group included MKN-45 (gastric adenocarcinoma), KNS62 (non-small cell lung carcinoma), RMG-I (ovarian carcinoma), and SK-MEL-24 (melanoma). The remaining six cell lines, namely AN3-CA (endometrial adenocarcinoma), H4 (glioma), LAMA-84 (chronic myeloid leukemia), SK-N-AS (neuroblastoma), SUIT-2 (pancreatic ductal adenocarcinoma), and 22Rv1 (prostate adenocarcinoma), were categorized as low GI.

We applied the same TF estimation analysis using informCNA to these two groups of samples. As shown in Figure R9, the low GI cancer group exhibited slightly reduced tumor DNA detection capacity. In DNA admixture samples with 0.2% TF, the low GI group showed a median estimated TF of 0% (IQR: 0.0-0.2) with an AUC of 0.64 for differentiating

samples with and without tumor DNA, demonstrating slightly lower sensitivity (48%) at 80% specificity. In contrast, the high GI group showed a higher median estimated TF of 0.1% (IQR: 0.0-0.2) and an improved AUC of 0.75 with 61% sensitivity at 80% specificity. Moreover, in the high GI group, DNA admixture samples with 0.1% TF exhibited a median estimated TF above 0%, further suggesting enhanced sensitivity in high GI cancer types. We have included the results as Extended Figure EV3 and discussed the results in the revised manuscript (pages 32 to 33, lines 908 to 917).

Figure R9. Sensitivities and specificities of cancer cell lines from cancer types with high and low genomic instability.

9. Also the purity of the reference sample is an important parameter. In figure 5, TF is shown to be robust even with low tumor content reference samples. It would be important with some more precise reporting on the needed tumor content. This is important since a frozen biopsies or baseline samples with detectable tumor content is often a limitation.

Response: We thank the reviewer for highlighting the importance of tumor purity in reference samples. We have now focused on low-purity tumor samples ($n = 6$) with purity level below 15%. As shown in Figure R10, we used low-purity tumor samples (median: 9.2%; range: 4.6-10.9%) as references and compared them with their higher-purity counterparts (median: 49.9%; range: 7.8-82.5%). Although the high-purity references showed more than a fivefold higher tumor fraction than the low-purity ones ($P = 0.03$, Wilcoxon signed-rank test, Figure R10A), the estimated TFs of their corresponding query samples remained highly consistent, with Pearson's $r = 0.97$ ($P < 0.001$). These results further support the robustness of informCNA in analyzing low purity samples. We have included the analysis in the revised manuscript (page 33, lines 936 to 940).

Figure R10. ctDNA detection using references with low tumor purity biopsies. **(A)** Tumor purity of low-purity biopsies (median: 9.2%; range: 4.6 to 10.9%) and their high-purity compartments (median: 49.9%; range: 7.8 to 82.5%). **(B)** Correlation between TFs estimated by informCNA using either the high- or low-purity biopsy as the reference sample in same patient.

10. The analysis done for the use of high-TF baseline (pre-treatment) plasma samples instead of tumor sample as a high TF reference is very limited. It can be a good addition to the study, if the authors run some additional analyses to validate their tool using pre-treatment plasma. Also it is not clear if the user wants to use pre-treatment plasma instead of tumor as reference, how high should be the TF in the baseline plasma? To specify this limit, more analyses are needed.

Response: We thank the reviewer for highlighting the importance of pre-treatment plasma to ctDNA detection in a plasma-only context. Clinically, tumor tissues generally have higher TF than plasma samples. Therefore, in terms of TF, using tumor tissue as a reference is preferable. However, in many scenarios, tissue biopsies are difficult to obtain, while minimally invasive plasma liquid biopsies can serve as a surrogate reference. This highlights the importance of extending the reference to baseline plasma samples.

We regret that no additional clinical data with baseline plasma showing varying TF is available for further analysis. Nevertheless, our existing analyses may address the reviewer's question about the lowest TF in baseline plasma for robust analysis. Our analyses between tumor and plasma samples suggest a high concordance between CNA segments identified from tumor tissue and plasma samples. Therefore, the minimum TF required in baseline plasma to yield informative CNAs should be comparable to that required in tumor tissue. As demonstrated in our response to Review's Comment 9, low-TF samples with a median TF of 9.2% (range: 4.6–10.9%) produced highly robust results when compared with their high-TF counterparts. As discussed in the blow Comment 12, before applying informCNA, we recommend user ensuring that the reference sample has an adequate TF and displays detectable CNAs using tumor-naïve tools

11. MRDetect seem to run sub-optimal in the analysis since it does not detect samples with TF 1-10% (Figure 4D). In the introduction a sensitivity of MRDetect of 0.005% TF is quoted. One difference to MRDetect seems to be that they use CNVkit for this analysis that might explain this?

Response: We agree with the reviewer's interpretation. In the benchmarking section comparing tumor-naïve and tumor-informed assays, we generated CNA calls from sWGS data of tumor samples without matched normal controls and applied all four assays to the sWGS data of plasma cfDNA samples. A potential limitation in detection sensitivity in MRDetect-CNA arises from the prior knowledge of tumor CNAs. For example, in determining tumor CNVs, Dan Landau's team used NBIC-seq (Xi et al., 2010) in MRDetect-CNA and Sequenza (Favero et al., 2014) in MRD-EDGE^{CNV}. Both methods require high-depth sequencing of tumor and matched normal tissues, resulting in single base-pair resolution CNA calls. Our intent in this comparison was not to equate the two approaches directly, but rather to benchmark their relative detection performance under certain use scenarios. We have clarified this point more explicitly in the revised manuscript (page 23, lines 611 to 615) to emphasize that our benchmarking highlights methodological difference and use-case contexts rather than suggesting a one-to-one comparison.

12. From the methods section, it seems that ACE tool was used for the estimation of tumor purity. If the LLOD of this tool is 5% (according to the manuscript), doesn't this affect the analyses of reference tumors with TF below 5%?

Response: We thank the reviewer for raising this point. Strategically, informCNA relies on prior knowledge of CNAs present in the reference sample. If the tumor fraction in the reference samples is below 5%, the estimation of TFs and CNA segmentation may be unreliable. Therefore, we recommend that one should ensure the reference sample has an adequate TF before applying informCNA. We have clarified this point in the revised README.md file and the revised manuscript (page 15, lines 433 to 435).

13. What does similarity score above 100 mean (Fig. S1 (A))? Is not 100 the max?

Response: The Similarity Score represents the TF ratio between reference and query samples based on genome-wide copy number profiles. This ratio can exceed 100% when the query sample is considered to have a higher TF than the reference sample. We have clarified it in the revised manuscript (page 32, lines 900 to 901).

14. Fig. S1(B): what is the reference here for the similarity score?

Response: In Figure S1(B), we performed the Copy Number Similarity Analysis between replicates under the same experimental condition. In these pairwise comparisons, each replicate, in turn, served as the reference for comparison with the others. We have clarified it in the revised manuscript (page 32, lines 903 to 904).

15. Estimation of run time of the tool? Comparison with other tools like MRDetect/MRD-EDGE or some comment on it?

Response: The main difference in strategy between informCNA and MRDetect/MRD-EDGE lies in how copy number aberrations are processed. For example, MRDetect-CNA requires precise copy number calling derived from high-depth sequencing of both tumor and matched normal tissue, focusing on single-base resolution and accurate coverage at each genomic position. This approach generates very large intermediate files (typically several gigabases). In contrast, informCNA focuses on genome-wide copy number changes using 100 kb bins to provide an overview of the entire genome, allowing for much faster computation. For example, using an AMD EPYC 7302 16-Core Processor, it takes ~8 minutes to profile the coverage of 1x sequencing data with the *cov.py* script, ~55 seconds for bin normalization and segmentation using the *seg.py* script, and ~12 seconds to compare reference and query coverage profiles using the *cmp.py* script. In comparison, MRDetect-CNA requires around 40 minutes to process sample coverage and an additional ~10 minutes to generate z-scores.

16. System's requirement to run the pipeline (memory and CPU)?

Response: informCNA is lightweight in both memory and computation, consuming less than 1GB of RAM and can be run on a standard laptop CPU. It mainly employs three Python scripts to generate results:

1. *cov.py* - Count genome coverage: Optimized to use <1 GB of memory, generating a genome-wide copy number profile (~12 MB).
2. *seg.py* - Normalization and segmentation: Uses <1 GB of memory to produce a genome-wide copy number segmentation result (~20 MB).
3. *cmp.py* - Copy Number Similarity and Significance Analysis: Uses <1 GB of memory to generate PDF outputs (~40 KB) and text results (~1 KB).

17. A list of abbreviations would be useful for the reader

Response: We have included a list of abbreviations in the revised manuscript (pages 23 to 24, lines 612 to 628).

18. MRDetect should be MRDetect

Response: We have corrected the typo.

Referee #3 (Remarks for Author):

This manuscript describes informCNA, a tumor-informed copy number analysis framework for ctDNA detection from shallow WGS. The approach leverages patient-specific CNA profiles to improve sensitivity, claiming a limit of detection of 0.2% TF while maintaining specificity. The work is timely and relevant, addressing an important need for cost-effective, scalable ctDNA monitoring methods. Benchmarking against established CNA tools and clinical validation in ovarian, breast, and melanoma cohorts underscore its translational potential. Please find below some suggestions to further strengthen the paper and its claims.

1. Validation Design and Potential Confounders

The *in silico* dilution mixes cell-line WGS into cfDNA from healthy donors. This may introduce a confounder, as tumor tissue DNA and plasma cfDNA differ substantially in fragmentation patterns (an effect demonstrated previously by this group). It is not clear how this impacts informCNA's reported sensitivity.

Response: We thank the reviewer for the question. Tissue gDNA fragments generated by sonication generally have random fragment ends and provide more uniform genomic coverage. In contrast, plasma cfDNA typically shows slightly lower coverage in open chromatin regions (e.g., gene promoter regions within 1 kb flanking transcription start sites). As shown by Sun *et al.* (Genome Res., 2019), cfDNA fragment-end density does not show a significant reduction near open chromatin regions, suggesting that cfDNA fragment numbers are not strongly biased in these areas and that the reduction in coverage is mainly attributable to shorter fragment lengths (Snyder *et al.* 2016). Since copy number analysis relies on read counts (i.e., the number of DNA fragments) within (100 kb) bins, we believe that informCNA should not be substantially biased in the sWGS setting.

Beyond the fragmentomic differences between tissue gDNA and cfDNA, the biological differences between ctDNA and normal cfDNA may have a greater impact on informCNA performance. Tumor-derived ctDNA exhibits distinct fragmentation patterns, including shorter fragment lengths and increased randomness in fragment end positioning. Selective analysis of short plasma cfDNA fragments (<150 bp) has been shown to enhance ctDNA detection by enriching for tumor-derived DNA fraction, with approximately twofold median enrichment (Mouliere *et al.*, 2018). In the previous manuscript, to assess informCNA performance in specific TF levels, we generated *in silico* dilution series by mixing defined amounts of artificially fragmented genomic DNA (gDNA) from cancer cell lines with known amounts of normal cfDNA. Therefore, the degree of tumor DNA enrichment could not be directly assessed in this *in silico* synthesized dataset. Based on the reviewer's suggestion, to demonstrate ctDNA enrichment by size selection to enhance the performance of informCNA, we have now applied *in silico* size selection to plasma cfDNA from the breast cancer cohort.

As shown in Figure R11, we compared TF estimates derived from overall plasma cfDNA without size selection to those from plasma cfDNA following size selection for short DNA molecules (<150 bp). informCNA detected ctDNA in 18 out of 30 samples, including 12/14 time points in patient P-IV-01, 7/7 time points in patient P-IV-02, and 1/9 time points in patient P-IV-03. Notably, in patient P-IV-02, informCNA identified ctDNA at two additional

time points compared with overall plasma DNA analysis, without introducing false negatives in all time points. Further comparison of TF estimates with and without size selection revealed a strong correlation with a slope of 1.55 (Figure R11C). It indicates an average 55% increase in ctDNA fraction after *in silico* size selection, consistent with previous findings (Mouliere *et al.*, 2018). We have included this analysis in the revised manuscript (pages 21 to 22, lines 560 to 572; page 34, lines 946 to 949).

Figure R11. informCNA results comparing analyses with and without *in silico* selection of short plasma cfDNA fragments.

Similarly, the *in vitro* dilution experiments are compelling, but details of the positivity cutoff ($\geq 0.2\%$ TF plus SVM positivity) could be explained more explicitly in the Results and figure legends.

Response: We thank the reviewer for the valuable suggestion. We have now explained it more explicitly in the revised manuscript (page 30, lines 834 to 838).

2. Lower Limit of Detection

The authors highlight 0.2% TF detection as a key advance. However, at this level the AUC is only 0.67 with ~50% sensitivity at 80% specificity. Is this sufficient for clinical use? I suggest tempering the claim or contextualizing it more explicitly against clinical requirements for recurrence monitoring.

Response: We agree with the reviewer that higher specificity is clinically important. We have now clarified that LLOD of informCNA is defined at 50% sensitivity. In the previous, manuscript, we also report sensitivities across multiple specificity levels (80%, 85%, 90%, and 95%) for detecting tumor DNA at different TFs, as shown in Figure 2D. For example, at 0.3% TF, informCNA achieved 34% sensitivity at 90% specificity, and 52% sensitivity at 85% specificity. In clinical applications, such as ctDNA detection in prognostic settings, greater emphasis may be placed on sensitivity rather than specificity. In this context, informCNA provides sensitivity and specificity tailored to the TF of the query sample.

In the clinical application of informCNA, as demonstrated in the ovarian cancer cohort (Figure EV6), TFs estimated by informCNA showed a stronger correlation with CA-125 levels than those estimated by ichorCNA across all 18 patients. Notably, in one patient (21521), the manually curated ichorCNA TF at the second plasma time point showed a decreasing trend, whereas both the CA-125 levels and the informCNA-estimated TF increased, further supporting the clinical relevance of informCNA.

informCNA has a LLOD of ~1000 ppm. In contrast, Black et al. (Nat Med 2025) and Black et al. (Cell 2025) demonstrated that a ctDNA level of 80 ppm both in pre- and postoperative is highly prognostic for risk stratification in early stage lung cancer. We argue that informCNA, despite its limited LLOD, may still be useful for clinically tracking patients with advanced-stage disease in a cost-efficient and frequent manner, and may aid in prompt treatment recommendations. Nevertheless, prospective validation in a larger cohort will be required to confirm its clinical utility.

3. Copy Number Landscape Dependence

Performance likely depends on the breadth and amplitude of CNAs in the reference tumor. Cancers that are copy-quiet may be challenging to monitor, and the differential contribution of copy gains vs losses is not explored. The rationale for placing extra weight on focal lesions is mentioned but not justified with results. A stratified analysis of performance by CNA burden, or at least a discussion, would strengthen the paper.

Response: We appreciate the valuable insight from the reviewer. informCNA relies on the copy number landscape of the reference sample to guide ctDNA detection in query samples. Thus, it tends to perform better in cancer types with high genomic instability, such as ovarian cancer, and may be more limited in copy-quiet cancer types. As shown in Figure R9 and in our response to Reviewer #2 Comment 8, genomic instability can influence assay sensitivity: tumors with high genomic instability (GI) facilitate ctDNA detection in follow-up plasma samples, achieving 61% sensitivity at 80% specificity at 0.2% TF, whereas cancer types with lower GI demonstrated slightly reduced sensitivity, 48%, at the same specificity at same TF. We have now included the analysis presented in Figure R9 and expanded the discussion on this impact of genomic instability across cancer types, along with practical guidance for informCNA users, in the revised manuscript (page 34, lines 946 to 949).

4. Benchmarking Strategy

It is a strength that three methods (t-MAD, ichorCNA, MRDetect-CNA) were included. However, benchmarking appears to have been done only in clinical samples. Was there a reason these methods were not also applied to the *in silico* and *in vitro* series, which would have enabled a more controlled performance comparison?

Response: We agree with the reviewer that benchmarking across a larger number of samples would provide a more comprehensive comparison. In the previous manuscript, we first assessed the performance of informCNA using an *in silico* admixture dataset and validated it with *in vitro* mixtures of ctDNA and plasma DNA. Subsequently, we benchmarked different assays in an independent, real-world clinical dataset to demonstrate their potential clinical utility. This cohort consisted of 30 plasma samples from cancer patients, including 12

samples with IMAF below 0.1% (median: 0.003; range: 0.002-0.079), 8 samples with IMAF between 0.1% and 1% (median: 0.32; range: 0.14-0.88), and 10 samples with IMAF above 1% (median: 3.59; range: 1.56-13.90). This balanced distribution of TFs enabled us to compare the performance of four different assays in terms of both sensitivity and specificity.

Based on the reviewer's suggestion, we first evaluated t-MAD and ichorCNA on the *in silico* dilution series at 1x coverage. As shown in Figure R12, because the dilution gradient was specifically designed for informCNA, only three tumor fraction categories (3%, 5% and 10%) were available to evaluate assay sensitivity.

Figure R12. ichorCNA, t-MAD and informCNA estimation results in dilution series.

To strengthen the benchmarking section, we therefore performed additional analyses on the previous clinical cohort to further characterize the performance of informCNA and other assays. We evaluated detection results at 1% and 0.1% IMAF (equivalent to approximately 2% and 0.2% TF, respectively). The cohort included 12 samples with IMAF below 0.1%, 8 samples between 0.1% and 1%, and 10 samples above 1%.

- t-MAD detected all 10 time points above 1%, but none in the 0.1% to 1% range.
- ichorCNA detected all 10 time points above 1%, and additionally identified two time points at 0.66% and 0.88%, detecting all ctDNA in plasma sample with IMAF above 0.5%.
- MRDetect-CNA detected 5 of 10 time points above 1%, and 5 time points in the 0.1%-1% range.
- informCNA detected all 18 time points above 0.1% IMAF.

We have included the analysis in the revised manuscript (pages 9 to 10, lines 262 to 280).

5. Impact of Sequencing Depth

The method is positioned as cost-effective due to low-coverage sWGS. However, sequencing costs are steadily declining, and some may consider deeper sequencing. Have the authors examined whether informCNA performance improves with higher depth? A sensitivity analysis would be helpful.

Response: We thank the reviewer for this question. As shown previously in Figure EV1, when we down-sampled tumor sequencing data to 0.1x coverage, we were still able to get approximately 98% of the copy number information, as quantified by the Similarity Score. Increasing the coverage to 0.2x and above yielded >99% consistency with tumor tissue sequenced at 5x coverage. As also discussed in our response to Reviewer #1 Comment 3.2,

increasing the sequencing depth improves the consistency of copy number profiling by less than 1% ($\geq 0.2x$) in terms of Similarity Score, providing limited benefit relative to the additional cost. The same principle also applies to query samples. We used tumor samples as an illustrative example because their CNAs provide a clear and quantifiable signal for benchmarking.

Importantly, the advantage of sWGS extend beyond sequencing cost. Lower cover substantially reduces data volume, which in turn shorten computational processing time and lowers analysis cost, resulting in a much faster turnaround time. This efficiency makes sWGS particularly well-suited for longitudinal monitoring and large-scale studies. We have clarified and strengthened this point in the revised manuscript (pages 13 to 14, lines 384 to 398).

6. Statistical Framework and Regularization

The Similarity Score relies on weighted least squares with LASSO regularization. The choice of penalty and its effect on detecting subtle CNA signals should be clarified. Since the approach aggregates many small deviations, how sensitive are results to regularization strength? A brief rationale and reference (e.g. DELFI, which also leverages subtle copy/fragmentomic signals) would help contextualize this choice.

Response: We thank the reviewer for the suggestions. In the Copy Number Similarity Analysis of informCNA, we used LASSO regularized least squares to robustly fit the genome-wide copy number profiles between reference and query samples. LASSO regularization mitigates perturbations in the copy number distribution arising from random or systematic noise (e.g. GC-content bias). The additional penalty term stabilizes the Similarity Score, leading to more conservative and robust estimates of TF with a reduced risk of overfitting. Consequently, this improves specificity and minimizes false positives by preventing overestimation of TF in healthy plasma samples.

In contrast, MRDetect-can aggregate read depth skews at patient-specific gain and loss segments across the genome for ctDNA detection (Zviran et al., 2020), conceptually similar to DELFI in that it accumulates subtle copy number signals. informCNA, on the other hand, leverages the entire genome-wide copy number landscape to quantify coverage similarity between reference and query samples. We have clarified this point and added additional description in the revised manuscript (page 20, lines 514 to 516).

7. Interpretation of Segment Correlation Figures

In the *in vitro* dilution series, the correlation structure (Fig. 3 lower panels) "lights up" as red with increasing TF. This seems to reflect segments beginning to correlate with each other, but the biological/statistical interpretation is not clear. Clarification in the text or legends would help readers.

Response: We thank the reviewer for the comment. In Figure 3, which shows the *in vitro* dilution series, segments are ranked by copy number in the reference sample (from lowest to highest, left to right), with the upper half colored yellow and the lower half colored green. The lower panel displays a p-value matrix from the segment comparison analysis, aligned with the same segment order. In this lower-triangular p-value matrix, each dot represents a

Dunn's test result, with colors transitioning from blue to red to indicate increasing statistical significance. The prominent red region in the lower-left corner corresponds to comparisons between the lowest to highest ranked segments, where the largest differences were observed. We have added the description to Fig.3 caption in the revised manuscript (page 30, lines 841 to 851).

Other Minor Concerns

- Figures: some multi-panel outputs (e.g. Fig. 3 significance plots) are difficult to follow without extended guidance.

Response: We have updated Fig. 3 caption with a revised layout and added description in the revised manuscript (page 30, lines 834 to 851).

- Discussion of potential confounding from clonal hematopoiesis CNAs in plasma-only mode is warranted.

Response: We thank the reviewer for the suggestion. Clonal hematopoiesis associated CNAs may confound analyses in the plasma-guided setting, and the persistent presence of non-tumor CNAs in plasma cfDNA may impair the performance of informCNA, resulting in an overestimation of TF due to false-positive signals. We have included this discussion in the revised manuscript (page 15, lines 425 to 429).

Reference

- Adalsteinsson VA, Ha G, Freeman SS, Choudhury AD, Stover DG, Parsons HA, Gydush G, Reed SC, Rotem D, Rhoades J, *et al* (2017) Scalable whole-exome sequencing of cell-free DNA reveals high concordance with metastatic tumors. *Nat Commun* 8: 1324
- Andor N, Maley CC, Ji HP. Genomic Instability in Cancer: Teetering on the Limit of Tolerance. *Cancer Res.* 2017 May 1;77(9):2179-2185
- Basyuni S, Heskin L, Degasperi A, Black D, Koh GCC, Chmelova L, Rinaldi G, Bell S, Grybowicz L, Elgar G, *et al* (2024) Large-scale analysis of whole genome sequencing data from formalin-fixed paraffin-embedded cancer specimens demonstrates preservation of clinical utility. *Nature Communications* 15: 7731
- Black, J.R.M., Bartha, G., Abbott, C.W. et al. Ultrasensitive ctDNA detection for preoperative disease stratification in early-stage lung adenocarcinoma. *Nat Med* 31, 70–76 (2025)
- Black JRM, Karasaki T, Abbott CW, et al. Longitudinal ultrasensitive ctDNA monitoring for high-resolution lung cancer risk prediction. *Cell.* 2025
- Diaz LA (2025) Cancer genetics in a tube of blood. *Nat Med*
- Favero F, Joshi T, Marquard AM, Birnbak NJ, Krzystanek M, Li Q, Szallasi Z, Eklund AC. Sequenza: allele-specific copy number and mutation profiles from tumor sequencing data. *Ann Oncol.* 2015 Jan;26(1):64-70
- Mouliere F, Chandrananda D, Piskorz AM, Moore EK, Morris J, Ahlborn LB, Mair R, Goranova T, Marass F, Heider K, Wan JCM, Supernat A, Hudcovova I, Gounaris I, Ros S, Jimenez-Linan M, Garcia-Corbacho J, Patel K, Østrup O, Murphy S, Eldridge MD, Gale D, Stewart GD, Burge J, Cooper WN, van der Heijden MS, Massie CE, Watts C, Corrie P, Pacey S, Brindle KM, Baird RD, Mau-Sørensen M, Parkinson CA, Smith CG, Brenton JD, Rosenfeld N. Enhanced detection of circulating tumor DNA by fragment size analysis. *Sci Transl Med.* 2018 Nov 7;10(466):eaat4921

- Poell, J. B. et al. ACE: Absolute copy number estimation from low-coverage whole-genome sequencing data. *Bioinformatics* 35, 2847–2849 (2019).
- Rickles-Young M, Tinoco G, Tsuji J, Pollock S, Haynam M, Lefebvre H, Glover K, Owen DH, Collier KA, Ha G, Adalsteinsson VA, Cibulskis C, Lennon NJ, Stover DG. Assay Validation of Cell-Free DNA Shallow Whole-Genome Sequencing to Determine Tumor Fraction in Advanced Cancers. *J Mol Diagn.* 2024 May;26(5):413-422.
- Scheinin I, Sie D, Bengtsson H, van de Wiel MA, Olshen AB, van Thuijl HF, van Essen HF, Eijk PP, Rustenburg F, Meijer GA, Reijneveld JC, Wesseling P, Pinkel D, Albertson DG, Ylstra B. DNA copy number analysis of fresh and formalin-fixed specimens by shallow whole-genome sequencing with identification and exclusion of problematic regions in the genome assembly. *Genome Res.* 2014 Dec;24(12):2022-32
- Steele CD, Abbasi A, Islam SMA, Bowes AL, Khandekar A, Haase K, Hames-Fathi S, Ajayi D, Verfaillie A, Dhami P, McLatchie A, Lechner M, Light N, Shlien A, Malkin D, Feber A, Proszek P, Lesluyes T, Mertens F, Flanagan AM, Tarabichi M, Van Loo P, Alexandrov LB, Pillay N. Signatures of copy number alterations in human cancer. *Nature.* 2022 Jun;606(7916):984-991
- Sauer, C. M. et al. Absolute copy number fitting from shallow whole genome sequencing data. *bioRxiv* (2021) doi:10.1101/2021.07.19.452658.
- Snyder MW, Kircher M, Hill AJ, Daza RM, Shendure J. Cell-free DNA Comprises an In Vivo Nucleosome Footprint that Informs Its Tissues-Of-Origin. *Cell.* 2016;164(1-2):57-68. doi:10.1016/j.cell.2015.11.050
- Sun K, Jiang P, Cheng SH, et al. Orientation-aware plasma cell-free DNA fragmentation analysis in open chromatin regions informs tissue of origin. *Genome Res.* 2019;29(3):418-427. doi:10.1101/gr.242719.118
- Talevich E, Shain AH, Botton T, Bastian BC. CNVkit: Genome-Wide Copy Number Detection and Visualization from Targeted DNA Sequencing. *PLoS Comput Biol.* 2016;12(4):e1004873
- Tie J, Cohen JD, Lahouel K, Lo SN, Wang Y, Kosmider S, Wong R, Shapiro J, Lee M, Harris S, et al (2022) Circulating Tumor DNA Analysis Guiding Adjuvant Therapy in Stage II Colon Cancer. *New England Journal of Medicine* 386: 2261-2272
- Wan JCM, Heider K, Gale D, Murphy S, Fisher E, Mouliere F, Ruiz-Valdepenas A, Santonja A, Morris J, Chandrananda D, et al (2020) ctDNA monitoring using patient-specific sequencing and integration of variant reads. *Sci Transl Med* 12: eaaz8084
- Widman AJ, Shah M, Frydendahl A, Halmos D, Khamnei CC, Øgaard N, Rajagopalan S, Arora A, Deshpande A, Hooper WF, et al (2024) Ultrasensitive plasma-based monitoring of tumor burden using machine-learning-guided signal enrichment. *Nat Med* 30: 1655-1666
- Xi R, Luquette J, Hadjipanayis A, Kim TM, Park PJ. BIC-seq: a fast algorithm for detection of copy number alterations based on high-throughput sequencing data. *Genome Biol.* 2010;11
- Zviran A, Schulman RC, Shah M, Hill STK, Deochand S, Khamnei CC, Maloney D, Patel K, Liao W, Widman AJ, et al (2020) Genome-wide cell-free DNA mutational integration enables ultra-sensitive cancer monitoring. *Nat Med* 26: 1114-1124

19th Dec 2025

Dear Dr. Zhao,

Thank you for submitting your revised study. We have now received the reports from the three referees. As you will see below, they are satisfied with the revisions, and I will therefore be able to accept your manuscript once the following editorial concerns are addressed:

1/ Manuscript text:

- Please remove the yellow highlights and indicate in track changes mode any new modification.
- Please correct the order and headings of the manuscript sections to: Abstract / The Paper Explained / Introduction / Results / Discussion / Methods / Data Availability / Acknowledgements / Disclosure and Competing Interests Statement / References / Main Figure Legends / Expanded View Figure Legends.
- Materials and Methods should be Methods.
- Please remove the Reagent and Tools table from the manuscript and upload it separately.
- Human samples: if applicable, please state details of authority granting ethics approval, include statements on informed consent and Helsinki declaration.
- List of Supplementary Materials section should be removed.
- Abbreviations section needs to be removed from the manuscript. Abbreviations should be defined in brackets after their first mention in the text, not in a list of abbreviations.
- Data and materials availability should be renamed Data Availability. Please note that only primary datasets produced in this study need to be listed here. Code Availability heading should be removed and the text underneath should be part of the section Data Availability. Please also remove "The underlying code for this study will be made available to qualified researchers on reasonable request from the corresponding author." As per our policy, datasets and computer code that are generated in the reported study should be listed in the Data Availability section and made publicly available.
- Acknowledgements: Please remove "For the purpose of open access, the authors have applied a Creative Commons Attribution (CC BY) license to the Author Accepted Manuscript." Funding should be part of Acknowledgements. 'Funding' heading should be removed; this funder appears to be missing in the submission system: Breast Cancer Now for funding as part of Program Funding to the Breast Cancer Now Toby Robins Research Centre (Grant Number CTR-Q5-Y3).
- References: Please note that the data citations are not tagged with the label "DATASET" in the reference list. In the main text, data citations are formatted as follows: "Data ref: Smith et al, 2001" or "Data ref: NCBI Sequence Read Archive PRJNA342805, 2017". In the Reference list, data citations must be labeled with "[DATASET]". A data reference must provide the database name, accession number/identifiers and a resolvable link to the landing page from which the data can be accessed at the end of the reference.

2/ Figures:

- Please make sure that all figures/figure panels are referenced in the text. Currently, callouts are missing for Figure 1C, Figure 3A, B, Figure 4F.
- You currently have a Table EV1 and a Table EV2: the correct nomenclature in all places needs to be Table EV1 for the first one and Dataset EV1 for the other table (Table EV2); "supplementary" should not be used; this nomenclature needs to be updated in source file names, titles in the system, legends, callouts in the manuscript; the legends need to be removed from the manuscript and each should be in the same Excel file (for Table EV1, the legend should be in the first row, for Dataset EV1, the legend should go in the other sheet/tab).
- Please address the queries from our data editors in the figure legends:
 1. Where possible, please provide exact p values in the legends of figures 4F, 5C, D; 6B, EV2 A, B; EV5A, B, D; EV7 B, EV8 B, D; EV9 B.
 2. Please note that the box plots need to be defined in terms of minima, maxima, centre, bounds of box and whiskers, and percentile in the legends of figures 1B, 5B, C; EV4 A, EV7 A
 3. Please note that information related to n is missing in the legends of figures 1B, 5B, C; EV1 B, EV2 A, B; EV4 A, EV7 A
 4. Please note that the error bars are not defined in the legends of figures EV1 B, EV2 A, B

3/ Please provide a complete author checklist, which you can download from our author guidelines. Please insert information in the checklist that is also reflected in the manuscript. The completed author checklist will also be part of the RPF.

4/ Thank you for providing Source Data. Please upload each folder (for each figure) separately. The text files should be removed. URL for deposited data needs to be in the Data Availability section.

5/ Thank you for providing a synopsis image and a thumbnail. Please resize them to 550 px wide x 300-600 px high, and 115 px wide x 70 px high, respectively.

6/ As part of the EMBO Publications transparent editorial process initiative (see our Editorial at

<http://embomolmed.embopress.org/content/2/9/329>), EMBO Molecular Medicine will publish online a Review Process File (RPF) to accompany accepted manuscripts.

This file will be published in conjunction with your paper and will include the anonymous referee reports, your point-by-point response and all pertinent correspondence relating to the manuscript. Let us know whether you agree with the publication of the RPF and as here, if you want to remove or not any figures from it prior to publication.

I look forward to receiving your revised manuscript.

Yours sincerely,

Lise Roth

***** Reviewer's comments *****

Referee #1 (Comments on Novelty/Model System for Author):

My assessment of the technical quality, novelty, medical impact and adequacy of the model system remain unchanged from my original review.

Referee #1 (Remarks for Author):

The authors have addressed my comments. The manuscript is significantly improved and, in my opinion, well-suited for publication at EMBO Mol Med.

Referee #2 (Remarks for Author):

Is suitable for publication

Referee #3 (Remarks for Author):

The authors have thoughtfully addressed all of my suggestions. I would like to congratulate the authors on this important contribution, and wish everyone a happy and healthy 2026!

The authors addressed the remaining editorial issues.

25th Feb 2026

Dear Dr. Zhao,

Thank you for addressing the last editorial issues. I am pleased to inform you that your manuscript is accepted for publication and is now being sent to our publisher to be included in the next available issue of EMBO Molecular Medicine.

Please note that due to its nature, the type of manuscript has been changed from 'Article' to 'Method'.

You may qualify for financial assistance for your publication charges - either via a Springer Nature fully open access agreement or an EMBO initiative. Check your eligibility: <https://link.springer.com/journal/44321/how-to-publish-with-us>

Yours sincerely,

Lise Roth

>>> Please note that it is EMBO Molecular Medicine policy for the transcript of the editorial process (containing referee reports and your response letter) to be published as an online supplement to each paper. If you do NOT want this, you will need to inform the Editorial Office via email immediately. More information is available here: <https://link.springer.com/partners/embo-press/editorial-policies#Peer%20review>